# Ensemble everything everywhere: Multi-scale aggregation for adversarial robustness

## ABSTRACT

Adversarial examples pose a significant challenge to the robustness, reliability and alignment of deep neural networks. We propose a novel, easy-to-use approach to achieving high-quality representations that lead to adversarial robustness through the use of multi-resolution input representations and dynamic self-ensembling of intermediate layer predictions. We demonstrate that intermediate layer predictions exhibit inherent robustness to adversarial attacks crafted to fool the full classifier, and propose a robust aggregation mechanism based on Vickrey auction that we call *CrossMax* to dynamically ensemble them. By combining multi-resolution inputs and robust ensembling, we achieve significant adversarial robustness on CIFAR-10 and CIFAR-100 datasets without any adversarial training or extra data, reaching an adversarial accuracy of $\approx$72% (CIFAR-10) and $\approx$48% (CIFAR-100) on the RobustBench AutoAttack suite ($L_\infty = 8/255$) with a finetuned ImageNet-pretrained ResNet152. This represents a result comparable with the top three models on CIFAR-10 and a +5 % gain compared to the best current dedicated approach on CIFAR-100. Adding simple adversarial training on top, we get $\approx$78% on CIFAR-10 and $\approx$51% on CIFAR-100, improving SOTA by 5 % and 9 % respectively and seeing greater gains on the harder dataset. We validate our approach through extensive experiments and provide insights into the interplay between adversarial robustness, and the hierarchical nature of deep representations. We show that simple gradient-based attacks against our model lead to human-interpretable images of the target classes as well as interpretable image changes. As a byproduct, using our multi-resolution prior, we turn pre-trained classifiers and CLIP models into controllable image generators and develop successful transferable attacks on large vision language models.

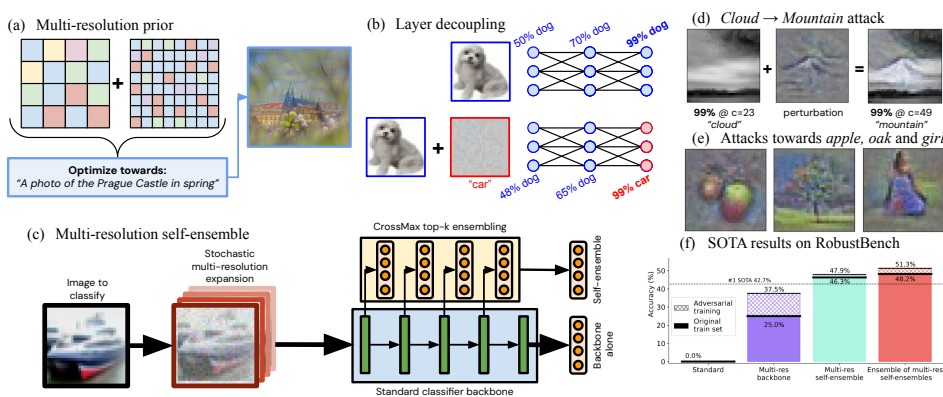

Figure 1: We use a multi-resolution decomposition (a) of an input image and a partial decorrelation of predictions of intermediate layers (b) to build a classifier (c) that has, by default, adversarial robustness comparable or exceeding state-of-the-art (f), even without any adversarial training. Optimizing inputs against it leads to interpretable changes (d) and images generated from scratch (e).

# 1 INTRODUCTION

Our objective is to take a step towards aligning the way machines perceive visual information – as expressed by the learned computer vision classification function – and the way people perceive visual information – as represented by the inaccessible, implicit human vision classification function. The significant present-day mismatch between the two is best highlighted by the existence of adversarial attacks that affect machine models but do not transfer to humans. Our aim is to develop a vision model with high-quality, natural representations that agree with human judgment not only under static perturbations, such as noise or dataset shift, but also when exposed to active, motivated attackers trying to dynamically undermine their accuracy. While adversarial robustness serves as our primary case study, the broader implications of this alignment extend to aspects such as interpretability, image generation, and the security of closed-source models, underscoring its importance.

Adversarial examples in the domain of image classification are small, typically human-imperceptible perturbations $P$ to an image $X$ that nonetheless cause a classifier, $f : X \rightarrow y$, to misclassify the perturbed image $X + P$ as a target class $t$ chosen by the attacker, rather than its correct, ground truth class. This is despite the perturbed image $X + P$ still looking clearly like the ground truth class to a human, highlighting a striking and consistent difference between machine and human vision (first described by Szegedy et al. (2013)). Adversarial vulnerability is ubiquitous in image classification, from small models and datasets (Szegedy et al., 2013) to modern large models such CLIP (Radford et al., 2021), and successful attacks transfer between models and architectures to a surprising degree (Goodfellow et al., 2015) without comparable transfer to humans. In addition, adversarial examples exist beyond image classification, for example in out-of-distribution detection, where otherwise very robust systems fall prey to such targeted attacks (Chen et al., 2021; Fort, 2022), and language modeling (Guo et al., 2021; Zou et al., 2023).

We hypothesize that the existence of adversarial attacks is due to the significant yet subtle mismatch between what humans do when they classify objects and how they learn such a classification in the first place (the *implicit* classification function in their brains), and what is conveyed to a neural network classifier explicitly during training by associating fixed pixel arrays with discrete labels (the learned machine classification function). It is often believed that by performing such a training we are communicating to the machine the implicit human visual classification function, which seems to be borne by their agreement on the training set, test set, behaviour under noise, and recently even their robustness to out-of-distribution inputs at scale (Fort et al., 2021b). We argue that while these two functions largely agree, the implicit human and learned machine functions are not *exactly* the same, which means that their mismatch can be actively exploited by a motivated, active attacker, purposefully looking for such points where the disagreement is large (for similar exploits in reinforcement learning see (Leike et al., 2017)). This highlights the difference between agreement on most cases, usually probed by static evaluations, and an agreement in all cases, for which active probing is needed.

In this paper, we take a step towards aligning the implicit human and explicit machine classification functions, and consequently observe very significant gains in adversarial robustness against standard attacks as a result of a few, simple, well-motivated changes, and without any explicit adversarial training. While, historically, the bulk of improvement on robustness metrics came from adversarial training (Chakraborty et al., 2018), comparably little attention has been dedicated to improving the model backbone, and even less to rethinking the training paradigm itself. Our method can also be easily combined with adversarial training, further increasing the model's robustness cheaply. Beyond benchmark measures of robustness, we show that if we optimize an image against our models directly, the resulting changes are human interpretable.

We operate under what what we call the **Interpretability-Robustness Hypothesis:** *A model whose adversarial attacks typically look human-interpretable will also be adversarially robust.* The aim of this paper is to support this hypothesis and to construct first versions of such robust classifiers, without necessarily reaching their peak performance via extensive hyperparameter tuning.

Firstly, inspired by biology, we design an active adversarial defense by constructing and training a classifier whose input, a standard $H \times W \times 3$ image, is stochastically turned into a $H \times W \times (3N)$ channel-wise stack of multiple downsampled and noisy versions of the same image. The classifier itself learns to make a decision about these $N$ versions *at once*, mimicking the effect of microsaccades in the human (and mammal) vision systems. Secondly, we show experimentally that hidden layer features of a neural classifier show significant de-correlation between their representations under

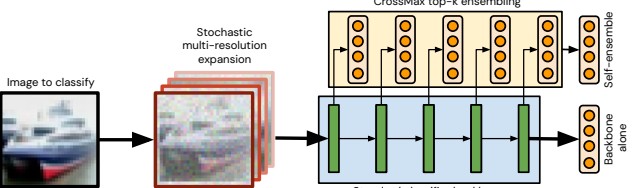

Figure 2: Combining channel-wise stacked augmented and down-sampled versions of the input image with robust intermediate layer class predictions via *CrossMax* self-ensemble. The resulting model gains a considerable adversarial robustness without any adversarial training or extra data.

adversarial attacks – an attack fooling a network to see a *dog* as a *car* does not fool the intermediate representations, which still see a *dog*. We aggregate intermediate layer predictions into a self-ensemble dynamically, using a novel ensembling technique that we call a *CrossMax* ensemble. Thirdly, we show that our Vickrey-auction-inspired *CrossMax* ensembling yields very significant gains in adversarial robustness when ensembling predictors as varied as 1) independent brittle models, 2) predictions of intermediate layers of the same model, 3) predictions from several checkpoints of the same model, and 4) predictions from several self-ensemble models. We use the last option to gain $\approx 5\%$ in adversarial accuracy at the $L_\infty = 8/255$ RobustBench's AutoAttack on top of the best models on CIFAR-100. When we add light adversarial training on top, we outperform current best models by $\approx 5\%$ on CIFAR-10, and by $\approx 9\%$ on CIFAR-100, showing a promising trend where the harder the dataset, the more useful our approach compared to brute force adversarial training (see Figure 6).

## 2 KEY OBSERVATIONS AND TECHNIQUES

In this section we will describe the three key methods that we use in this paper. In Section 2.1 we introduce the idea of multi-resolution inputs, in Section 2.2 we introduce our robust *CrossMax* ensembling method, and in Section 2.3 we showcase the de-correlation between adversarially induced mistakes at different layers of the network and how to use it as an active defense.

### 2.1 THE MULTI-RESOLUTION PRIOR

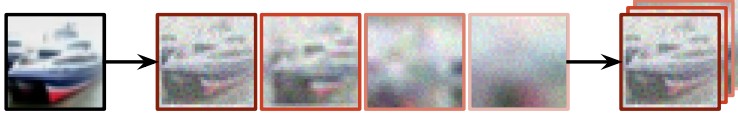

Figure 3: An image input being split into $N$ progressively lower resolution versions that are then stacked channel-wise, forming a $3N$-channel image input to a classifier.

Drawing inspiration from biology, we use multiple versions of the same image at once, down-sampled to lower resolutions and augmented with stochastic jitter and noise. We train a model to classify this channel-wise stack of images simultaneously. We show that this by default yields gains in adversarial robustness without any explicit adversarial training.

**Classifying many versions of the same object at once.** The human visual system has to recognize an object, e.g. a *cat*, from all angles, distances, under various blurs, rotations, illuminations, contrasts and similar such transformations that preserve the semantic content of whatever a person is looking at while widely changing the "pixel" values of the image projected on the retina.

A classification decision is not performed on a single frame but rather on a long stream of such frames that come about due to changing physical conditions under which an object is viewed as well as the motion of the eyes and changing properties of the retina (resolution, color sensitivity) at a place where the object is projected. We hypothesize that this is a key difference between the human visual system and a standard approach to image classification, where still, high-resolution frames

are associated with discrete labels. We believe that bridging this gap will lead to better alignment between the implicit human classification function, and the explicit machine classification function.

Augmentations that preserve the semantic content of images while increasing their diversity have historically been used in machine learning, for an early example see (LeCun et al., 1998). However, typically, a particular image $X$ appears in a single pass through the training set (an *epoch*) a single time, in its augmented form $X'$. The next occurrence takes place in the following epoch, with a different augmentation $X''$. In (Havasi et al., 2021), multiple images are fed into the network at once through independent subnetworks. In (Fort et al., 2021a), the same image $X$ is augmented $N$ times within the same batch, leading to faster training and higher final performance, likely due to the network having to learn a more transformation-invariant notion of the object at once. In this paper, we take this process one step further, presenting different augmentations as additional image channels *at the same time*. This can be viewed as a very direct form of ensembling.

**Biological eye saccades.** Human eyes (as well as the eyes of other animals with foveal vision) perform small, rapid, and involuntary jitter-like motion called *microsaccades* (cf. (Martinez-Conde et al., 2004) for details). The amplitude of such motion ranges from approximately 2 arcminutes to 100 arcminutes. In the center of the visual field where the human eye has the highest resolution, it is able to resolve up to approximately 1 arcminute. That means that even the smallest microsaccade motion moves the image projected on the retina by at least one pixel in amplitude. The resolution gradually drops towards the edges of the visual field to about 100 arcminutes (Wandell, 1995). Even there the largest amplitude macrosaccades are sufficient to move the image by at least a pixel. The standard explanation is that these motions are needed to refresh the photosensitive cells on the retina and prevent the image from fading (Martinez-Conde et al., 2004). However, we hypothesize that an additional benefit is an increase in the robustness of the visual system. We draw inspiration from this aspect of human vision and add deterministically random jitter to different variants of the image passed to our classifier. Apart from the very rapid and small amplitude microsaccades, the human eye moves around the visual scene in large motions called *macrosaccades* or just *saccades*. Due to the decreasing resolution of the human eye from the center of the visual field, a particular object being observed will be shown with different amounts of blur.

**Multi-resolution input to a classifier.** We turn an input image $X$ of full resolution $R \times R$ and 3 channels (RGB) into its $N$ variations of different resolutions $r \times r$ for $r \in \rho$. For CIFAR-10 and CIFAR-100, we are (arbitrarily) choosing resolutions $\rho = \{32, 16, 8, 4\}$ and concatenating the resulting image variations $\text{rescale}_R (\text{rescale}_r(X))$ channel-wise to a $R \times R \times (3|\rho|)$ augmented image $\bar{X}$. This is shown in Figure 3. Similar approaches have historically been used to represent images, such as Gaussian pyramids introduced in (Burt & Adelson, 1983). To each variant we add 1) random noise both when downsampled and at the full resolution $R \times R$ (in our experiments of strength 0.1 out of 1.0), 2) a random jitter in the $x - y$ plane ($\pm 3$ in our experiments), 3) a small, random change in contrast, and 4) a small, random color-grayscale shift. This can also be seen as an effective reduction of the input space dimension available to the attacker, as discussed in (Fort, 2023).

## 2.2 *CrossMax* ROBUST ENSEMBLING

**Robust aggregation methods, Vickrey auctions and load balancing.** The standard way of ensembling predictions of multiple networks is to either take the mean of their logits, or the mean of their probabilities. This increases both the accuracy as well as predictive uncertainty estimates of the ensemble (Lakshminarayanan et al., 2017; Ovadia et al., 2019). Such aggregation methods are, however, susceptible to being swayed by an outlier prediction by a single member of the ensemble or its small subset. This produces a single point of failure. The pitfalls of uncertainty estimation and ensembling have been highlighted in (Ashukha et al., 2021), while the effect of ensembling on the learned classification function was studied by Fort et al. (2022).

With the logit mean in particular, an attacker can focus all their effort on fooling a *single* network's prediction strongly enough towards a target class $t$. Its high logit can therefore dominate the full ensemble, in effect confusing the aggregate prediction. An equivalent and even more pronounced version of the effect would appear were we to aggregate by taking a `max` over classifiers per class. The calibration of individual members vs their ensemble is theoretically discussed in (Wu & Gales, 2021).

Our goal is to produce an aggregation method that is robust against an *active* attacker trying to exploit it, which is a distinct setup from being robust against e.g. untargeted perturbations. In fact, methods very robust against out-of-distribution inputs (Fort et al., 2021b) are still extremely brittle against *targeted* attacks (Fort, 2022). Generally, this observation, originally stated as *"Any observed statistical regularity will tend to collapse once pressure is placed upon it for control purposes"* in Goodhart (1981), is called *Goodhart's law*, and our goal is to produce an anti-Goodhart ensemble.

We draw our intuition from *Vickrey auctions* (Wilson, 1977) which are designed to incentivize truthful bidding. Viewing members of ensembles as individual bidders, we can limit the effect of wrong, yet overconfident predictions by using the $2^{\text{nd}}$ highest, or generally $k^{\text{th}}$ highest prediction per class. This also produces a cat-and-mouse-like setup for the attacker, since *which* classifier produces the $k^{\text{th}}$ highest prediction for a particular class changes dynamically as the attacker tries to increase that prediction. A similar mechanism is used in balanced allocation (Azar et al., 1999) and specifically in the *k random choices* algorithm for load balancing (Mitzenmacher et al., 2001).

Our *CrossMax* aggregation works a follows: For logits $Z$ of the shape $[B, N, C]$, where $B$ is the batch size, $N$ the number of predictors, and $C$ the number of classes, we first subtract the max per-predictor $\max(Z, \text{axis} = 1)$ to prevent Goodhart-like attacks by shifting the otherwise-arbitrary overall constant offset of a predictor's logits. This prevents a single *predictor* from dominating. The second, less intuitive step, is subtracting the per-class max to encourage the winning class to win via a consistent performance over many predictors rather than an outlier. This is to prevent any *class* from spuriously dominating. We aggregate such normalized logits via a per-class `topk` function for our self-ensembles and `median` for ensembles of equivalent models, as shown in Algorithm 1.

---

**Algorithm 1** CrossMax = An Ensembling Algorithm with Improved Adversarial Robustness

**Require:** Logits $Z$ of shape $[B, N, C]$, where $B$ is the batch size, $N$ the number of predictors, and $C$ the number of classes
**Ensure:** Aggregated logits
1: $\hat{Z} \leftarrow Z - \max(Z, \text{axis} = 2)$ {Subtract the max per-predictor over classes to prevent any predictor from dominating}
2: $\hat{Z} \leftarrow \hat{Z} - \max(\hat{Z}, \text{axis} = 1)$ {Subtract the per-class max over predictors to prevent any class from dominating}
3: $Y \leftarrow \text{median}(\hat{Z}, \text{axis} = 1)$ {Choose the median (or $k^{\text{th}}$ highest for self-ensemble) logit per class}
4: **return** $Y$

---

We use this aggregation for intermediate layer predictions (changing *median* to $top_3$) as well and see similar, transferable gains. We call this setup a *self-ensemble*.

## 2.3 ONLY PARTIAL OVERLAP BETWEEN THE ADVERSARIAL SUSCEPTIBILITY OF INTERMEDIATE LAYERS

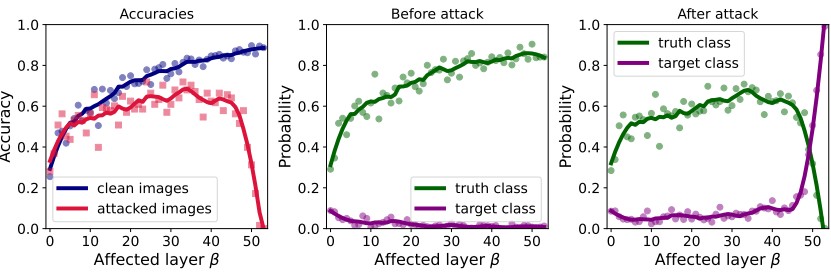

Figure 4: The impact of adversarial attacks ($L_\infty = 8/255$, 128 attacks) against the full classifier on the accuracy and probabilities at all intermediate layers for an ImageNet-1k pretrained ResNet152 finetuned on CIFAR-10 via trained linear probes.

A key question of both scientific and immediately practical interest is whether an adversarially modified image $X'$ that looks like the target class $t$ to a classifier $f : X \rightarrow y$ also has intermediate

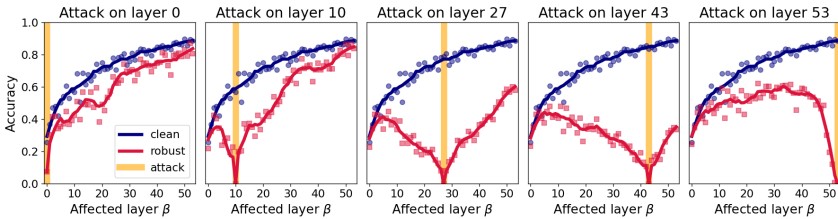

Figure 5: Transfer of adversarial attacks ($L_\infty = 8/255$, 512 attacks) against the activations of layer $\alpha$ on the accuracy of layer $\beta$ for $\alpha = 0, 10, 27, 43, 53$ on ImageNet-1k pretrained ResNet152 finetuned on CIFAR-10 via trained linear probes. Each panel shows the effect of designing a pixel-level attack to confuse the linear probe at a particular layer. For more details, see Figure 23.

layer representations that look like that target class. In (Olah et al., 2017), it is shown via feature visualization that neural networks build up their understanding of an image hierarchically starting from edges, moving to textures, simple patterns, all the way to parts of objects and full objects themselves. This is further explored by Carter et al. (2019). Does an image of a *car* that has been adversarially modified to look like a *tortoise* to the final layer classifier carry the intermediate features of the target class *tortoise* (e.g. the patterns on the shell, the legs, a tortoise head), of the original class *car* (e.g. wheels, doors), or something else entirely? We answer this question empirically.

To investigate this phenomenon, we fix a trained network $f : X \to y$ and use its intermediate layer activations $h_1(X), h_2(X), \cdots, h_L(X)$ to train separate trained linear probes (affine layers) that map the activation of the layer $l$ into classification logits $z_i$ as $g_i : h_i(X) \to y_i$. An image $X$ generates intermediate representations $(h_1, h_2, \ldots, h_L)$ that in turn generate $L$ different sets of classification logits $(z_1, z_2, \ldots, z_L)$. In Figure 4 we showcase this effect using an ImageNet-pretrained ResNet152 (He et al., 2015) finetuned on CIFAR-10. Images attacked to look like some other class than their ground truth (to the final layer classification) do not look like that to intermediate layers, as shown by the target class probability only rising in the very last layers (see Figure 4). We can therefore confirm that indeed the activations of attacked images do not look like the target class in the intermediate layers, which offers two immediate use cases: 1) as a warning flag that the image has been tempered with and 2) as an active defense, which is strictly harder.

This setup also allows us not only to investigate what the intermediate classification decision would be for an adversarially modified image $X'$ that confuses the network's final layer classifier, but also to generally ask what the effect of confusing the classifier at layer $\alpha$ would do to the logits at a layer $\beta$. The results are shown in Figure 5 for 6 selected layers to attack, and the full attack layer $\times$ read-out layer is show in Figure 23.

We find that attacks designed to confuse early layers of a network do not confuse its middle and late layers. Attacks designed to fool middle layers do not fool early nor late layers, and attacks designed to fool late layers do not confuse early or middle layers. In short, there seems to be roughly a 3-way split: early layers, middle layers, and late layers. Attacks designed to affect one of these do not generically generalize to others. We call this effect the *adversarial layer de-correlation*. This de-correlation allows us to create a *self-ensemble* from a single model, aggregating the predictions resulting from intermediate layer activations.

## 3 TRAINING AND EXPERIMENTAL RESULTS

In this section we present in detail how we combine the previously described methods and techniques into a robust classifier on CIFAR-10 and CIFAR-100. We start both with a pretrained model and finetune it, as well as with a freshly initialized model.

**Model and training details.** The pretrained models we use are the ImageNet (Deng et al., 2009) trained ResNet18 and ResNet152 (He et al., 2016). Our hyperparameter search was very minimal and we believe that additional gains are to be had with a more involved search easily. The only architectural modification we make is to change the number of input channels in the very first convolutional layer from 3 to $3N$, where $N$ is the number of channel-wise stacked down-sampled images we use as input. We also replaced the final linear layer to map to the correct number of classes

(10 for CIFAR-10 and 100 for CIFAR-100). Both the new convolutional layer as well as the final linear layer are initialized at random. The batch norm (Ioffe & Szegedy, 2015) is on for finetuning a pretrained model (although we did not find a significant effect beyond the speed of training).

We focused on the CIFAR-* datasets (Krizhevsky, 2009; Krizhevsky et al.) that comprise 50,000 $32 \times 32 \times 3$ images. We arbitrarily chose $N = 4$ and the resolutions we used are $32 \times 32$, $16 \times 16$, $8 \times 8$, $4 \times 4$ (see Figure 3). We believe it is possible to choose better combinations, however, we did not run an exhaustive hyperparameter search there. The ResNets we used expect $224 \times 224$ inputs. We therefore used a `bicubic` interpolation to upsample the input resolution for each of the 12 channels independently. To each image (the $32 \times 32 \times 3$ block of RGB channels) we add a random jitter in the $x - y$ plane in the $\pm 3$ range. We also add a random noise of standard deviation 0.2 (out of 1.0). All training is done using the `Adam` (Kingma & Ba, 2015) optimizer at a flat learning rate $\eta$ that we always specify. Optimization is applied to all trainable parameters and the batch norm is turned on in case of finetuning, but turned off for training from scratch. Linear probes producing predictions at each layer are just single linear layers that are trained on top of the pre-trained and frozen backbone network, mapping from the number of hidden neurons in that layer (flattened to a single dimension) to the number of classes (10 for CIFAR-10 and 100 for CIFAR-100). We trained them using the same learning rate as the full network for 1 epoch each.

**Adversarial vulnerability evaluation.** To make sure we are using as strong an attack suite as possible to measure our networks' robustness and to be able to compare our results to other approaches, we use the `RobustBench` (Croce et al., 2020) library and its `AutoAttack` method, which runs a suite of four strong, consecutive adversarial attacks on a model in a sequence and estimates its adversarial accuracy (e.g. if the attacked images were fed back to the network, what would be the classification accuracy with respect to their ground truth classes). For faster evaluation during development, we used the first two attacks of the suite (APGD-CE and APGD-T) that are particularly strong and experimentally we see that they are responsible for the majority of the accuracy loss under attack. For full development evaluation (but still without the `rand` flag) we use the full set of four tests: APGD-CE, APGD-T, FAB-T and SQUARE. Finally, to evaluate our models using the hardest method possible, we ran the `AutoAttack` with the `rand` flag that is tailored against models using randomness. The results without adversarial training are shown in Table 1 and with adversarial training at Table 2. The visual representation of the results is presented in Figure 6.

Table 1: *Randomized* (strongest) RobustBench AutoAttack adversarial attack suite results at the $L_\infty = 8/255$ strength. In this table we show the results of attacking our multi-resolution ResNet152 models finetuned on CIFAR-10 and CIFAR-100 from an ImageNet pretrained state without any adversarial training or extra data for 20 epochs with Adam at $\eta = 3.3 \times 10^{-5}$. We use our *CrossMax* ensembling on the model itself (self-ensemble), the final 3 epochs (3-ensemble), and on self-ensembles from 3 different runs (3-ensemble of self-ensembles). We also include results for a ResNet18 trained from *scratch* on CIFAR-10. Additional adversarial training helps, as shown in Table 2.

| Dataset | Adv. train | Model | Method | # | Test acc | rand AutoAttack $L_\infty = 8/255$ (%) | | |
|---|---|---|---|---|---|---|---|---|
| | | | | | | Adv acc | APGD CE$\rightarrow$ | APGD DLR |
| CIFAR-10 | $\times$ | ResNet18* | Self-ensemble | 1024 | 76.94 | 64.06 | 51.56 | 44.53 |
| CIFAR-10 | $\times$ | ResNet152 | Multires backbone | 128 | 89.17 | 41.44 | 32.81 | 21.88 |
| CIFAR-10 | $\times$ | ResNet152 | Self-ensemble | 128 | 87.14 | 53.12 | 50.00 | 43.75 |
| CIFAR-10 | $\times$ | ResNet152 | 3-ensemble of self-ensembles | 128 | 90.20 | **71.88** | 68.75 | 68.75 |
| CIFAR-10 | $\checkmark$ | [3] | SOTA #1 | | | 73.71 | | |
| CIFAR-100 | $\times$ | ResNet152 | Multires backbone | 128 | 65.70 | 25.00 | 21.88 | 13.28 |
| CIFAR-100 | $\times$ | ResNet152 | Self-ensemble | 512 | 65.71 | **46.29** $\pm 2.36$ | 34.77 $\pm 2.09$ | 30.08 $\pm 2.13$ |
| CIFAR-100 | $\times$ | ResNet152 | 3-ensemble of self-ensembles | 512 | 67.71 | **48.16** $\pm 2.65$ | 40.63 $\pm 2.11$ | 37.32 $\pm 1.98$ |
| CIFAR-100 | $\checkmark$ | [48] | SOTA #1 | | | 42.67 | | |

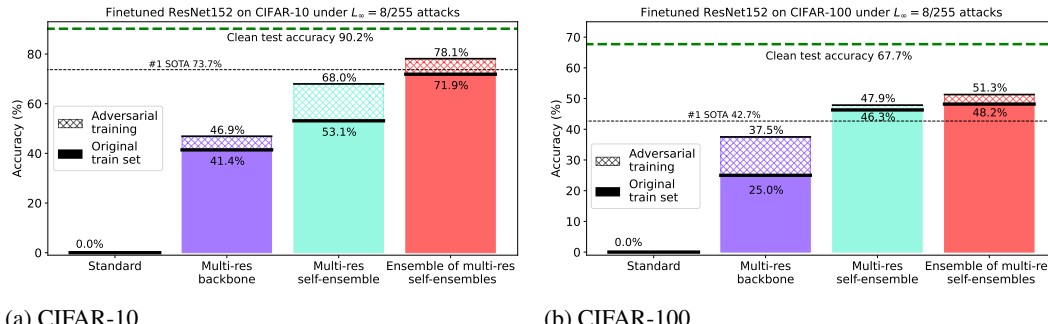

(a) CIFAR-10            (b) CIFAR-100

Figure 6: Adversarial robustness evaluation for finetuned ResNet152 models under $L_\infty = 8/255$ attacks of RobustBench AutoAttack (*rand* version = stronger against our models). On CIFAR-10, a CrossMax 3-ensemble of our self-ensemble multi-resolution models reaches #3 on the leaderboard, while on CIFAR-100 a 3-ensemble of our multi-resolution models is #1, leading by $\approx +5$ % in adversarial accuracy. When we add light adversarial training, our models surpass SOTA on CIFAR-10 by $\approx +5$ % and on CIFAR-100 by a strong $\approx +9$ %.

**Multi-resolution finetuning of a pretrained model.** In this section we discuss finetuning a standard pretrained model using our multi-resolution inputs. We demonstrate that this quickly leads to very significant adversarial robustness that matches and in some cases (CIFAR-100) significantly improves upon current best, dedicated approaches, without using any extra data or adversarial training. We see stronger gains on CIFAR-100 rather than CIFAR-10, suggesting that its edge might lie at harder datasets, which is a very favourable scaling compared to brute force adversarial training.

We show that we can easily convert a pre-trained model into a robust classifier without any data augmentation or adversarial training in a few epochs of standard training on the target downstream dataset. The steps we take are as follows: 1) Take a pretrained model (in our case ResNet18 and ResNet152 pretrained on ImageNet). 2) Replace the first layer with a fresh initialization that can take in $3N$ instead of 3 channels. 3) Replace the final layer with a fresh initialization to project to 10 (for CIFAR-10) or 100 (for CIFAR-100) classes. 4) Train the full network with a *small* (this is key) learning rate for a few epochs

We find that using a small learning rate is key, which could be connected to the effects described for example in Thilak et al. (2022) and Fort et al. (2020). While the network might reach a good clean test accuracy for high learning rates as well, only for small learning rates will it also get significantly robust against adversarial attacks, as shown in Figure 20.

In Table 1 we present our results of finetuning an ImageNet pretrained ResNet152 on CIFAR-10 and CIFAR-100 for 10 epochs at the constant learning rate of $3.3 \times 10^{-5}$ with Adam followed by 3 epochs at $3.3 \times 10^{-6}$. We find that even a simple 10 epoch finetuning of a pretrained model using our multi-resolution input results in a significant adversarial robustness. When using the strongest `rand` flag for models using randomized components in the RobustBench AutoAttack without any tuning against, we show significant adversarial robustness, as shown in Tab 1. On CIFAR-10, our results are comparable to the top three models on the leaderboard, despite never using any extra data or adversarial training. On CIFAR-100, our models actually lead by $+5\%$ over the current best model.

In Figure 6 we can see the gradual increase in adversarial accuracy as we add layers of robustness. First, we get to $\approx 40\%$ by using multi-resolution inputs. An additional $\approx 10\%$ is gained by combining intermediate layer predictions into a self-ensemble. An additional $\approx 20\%$ on top is then gained by using CrossMax ensembling to combining 3 different self-ensembling models together. Therefore, by using three different ensembling methods at once, we reach approximately $70\%$ adversarial accuracy on CIFAR-10. The gains on CIFAR-100 are roughly equally split between the multi-resolution input and self-ensemble, each contributing approximately half of the robust accuracy.

**Training from scratch.** We train a ResNet18 from scratch on CIFAR-10 as a backbone, and then train additional linear heads for all of its intermediate layers to form a CrossMax self-ensemble. We find that, during training, augmenting our input images $X$ with an independently drawn images $X'$ with a randomly chosen mixing proportion $p$ as $(1 - p)X + pX'$ increases the robustness of the

trained model. This simple augmentation technique is known as `mixup` and is described in Zhang et al. (2018). The results on the full `RobustBench` AutoAttack suite of attacks for CIFAR-10 are shown in Table 1 for self-ensemble constructed on top of the multi-resolution ResNet18 backbone (the linear heads on top of each layer were trained for 2 epochs with Adam at $10^{-3}$ learning rate).

**Adversarial finetuning.** Adversarial training, which adds attacked images with their correct, ground truth labels back to the training set, is a standard brute force method for increasing models' adversarial robustness. (Chakraborty et al., 2018) It is ubiquitous among the winning submissions on the RobustBench leader board, e.g. in Cui et al. (2023) and Wang et al. (2023). To verify that our technique does not only somehow replace the need for dedicated adversarial training, but rather that it can be productively combined with it for even stronger adversarial robustness, we re-ran all our finetuning experiments solely on adversarially modified batches of input images generated on the fly.

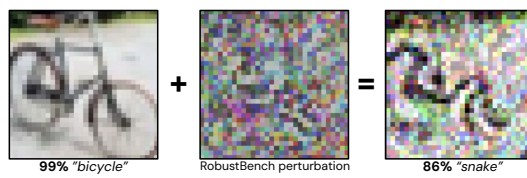

99% *"bicycle"* — RobustBench perturbation — 86% *"snake"*

Figure 7: An example of a $L_\infty = 64/255$ RobustBench AutoAttack on our model, changing a *bicycle* into a *snake* in an interpretable way.

For each randomly drawn batch, we used the single-step fast gradient sign method from Goodfellow et al. (2015) to *increase* its cross-entropy loss with respect to its ground truth labels. We used the $L_\infty = 8/255$ for all attacks. In Table 2 we show the detailed adversarial robustness of the resulting models. Figure 6 shows a comparison of the standard training and adversarial training for all models on CIFAR-10 and CIFAR-100. In all cases, we see an additive benefit of adversarial training on top of our techniques. In particular, for CIFAR-10 we outperform current SOTA by approximately 5 % while on CIFAR-100 and by approximately 9 % on CIFAR-100, which is a very large increase. The fact that our techniques benefit even from a very small amount of additional adversarial training (units of epochs of a single step attack) shows that our multi-resolution inputs and intermediate layer aggregation are a good prior for getting broadly robust networks.

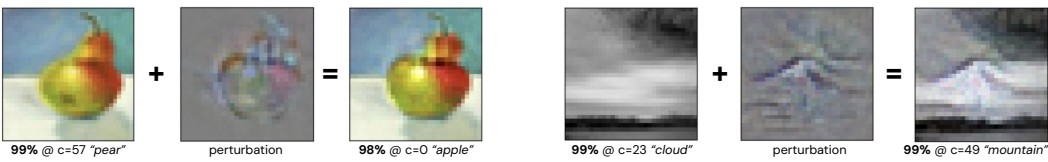

99% @ c=57 *"pear"* — perturbation — 98% @ c=0 *"apple"*         99% @ c=23 *"cloud"* — perturbation — 99% @ c=49 *"mountain"*

(a) *Pear* to *apple*                                       (b) *Cloud* to *mountain*

Figure 8: Examples of an adversarial attack on an image towards a target label. We use simple gradient steps with respect to our multi-resolution ResNet152 finetuned on CIFAR-100. The resulting attacks use the underlying features of the original image and make semantically meaningful, human-interpretable changes to it. Additional examples available in Figure 24.

**Visualizing attacks against multi-resolution models.** We wanted to visualize the attacks against our multi-resolution models. In Figure 8 we start with a test set image of CIFAR-100 (a *pear*, *cloud*, *camel* and *elephant*) and over 400 steps with SGD and $\eta = 1$ minimize the loss with respect to a target class (*apple*, *mountain*, *rabbit* and *dinosaur*). We allow for large perturbations, up to $L_\infty = 128/255$, to showcase the alignment between our model and the implicit human visual system classification function. In case of the *pear*, the perturbation uses the underlying structure of the fruit to divide it into 2 apples by adding a well-placed edge. The result-

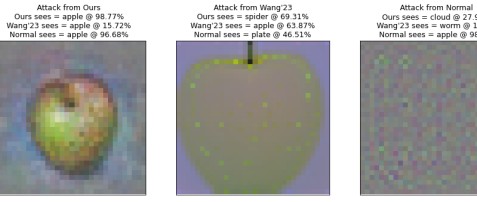

Figure 9: Examples of adversarial attacks on our multi-resolution ResNet152 finetuned on CIFAR-100 (left), the previous best model on CIFAR-100 $L_\infty = 8/255$ on RobustBench from Wang et al. (2023) (middle), and standard ResNet152 finetuned on CIFAR-100

ing image is very obviously an apple to a human as well as the model itself. In case of the cloud, its white color is repurposed by the attack to form the snow of a mountain, which is drawn in by a dark

sharp contour. In case of the elephant, it is turned into a dinosaur by being recolored to green and made spikier – all changes that are very easily interpretable to a human.

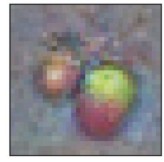 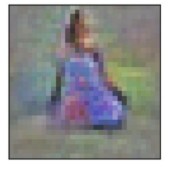 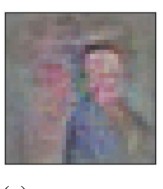 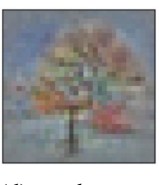 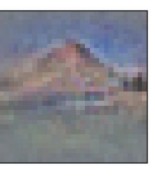

(a) *apple*          (b) *girl*          (c) *man*          (d) *maple*          (e) *mountain*

Figure 10: Examples of adversarial attacks on our multi-resolution ResNet152 finetuned on CIFAR-100. The attacks are generated by starting from a uniform image (128,128,128) and using gradient descent of the cross-entropy loss with SGD at $\eta = 1$ for 400 steps towards the target label. For standard models, these look like noise (Figure 9).

In Figure 10 we start with a uniform gray image of color (128, 128, 128) and by changing it to maximize the probability of a target class with respect to our model, we generate an image. The resulting images are very human-interpretable. This can be directly contrasted with the results in Figure 9 that one gets running the same procedure on a brittle model (noise-like patterns) and a current best, adversarially trained CIFAR-100 model ((Wang et al., 2023); suggestive patterns, but not real images). We also generated 4 examples per CIFAR-100 class for all 100 classes in Figure 26 to showcase that we do not cherrypick the images shown.

Figure 25 shows 6 examples of successfully attacked CIFAR-100 test set images for an ensemble of 3 self-ensemble models – our most adversarially robust model. When looking at the misclassifications caused, we can easily see human-plausible ways in which the attacked image can be misconstrued as the most probable target class. Figure 7 shows a successful $L_\infty = 64/255$ (much larger than the standard 8/255 perturbations) RobustBench AutoAttack on a test image of a *bicycle* converting it, in a human-interpretable way, to a *snake* by re-purposing parts of the bicycle frame as the snake body.

## 4 DISCUSSION AND CONCLUSION

In this paper, we introduced a novel approach to bridging the gap between machine and human vision systems. Our techniques lead to higher-quality, natural representations that improve the adversarial robustness of neural networks by leveraging multi-resolution inputs and a robust (self-)ensemble aggregation method we call CrossMax. Our method approximately matches state-of-the-art adversarial accuracy on CIFAR-10 and exceeds it on CIFAR-100 without relying on any adversarial training or extra data at all. When light adversarial training is added, it sets a new best performance on CIFAR-10 by $\approx 5\%$ and by a significant $\approx 9\%$ on CIFAR-100, taking it from $\approx 40\%$ to $\approx 50\%$. Key contributions of our work include: 1) Demonstrating the effectiveness of multi-resolution inputs as an active defense mechanism against adversarial attacks and a design principle for higher-quality, robust classifiers. 2) Introducing the CrossMax ensemble aggregation method for robust prediction aggregation. 3) Providing insights into the partial robustness of intermediate layer features to adversarial attacks. 4) Supporting the Interpretability-Robustness Hypothesis through empirical evidence. 5) Discovering a method to turn pre-trained classifiers and CLIP models into controllable image generators. 6) Generating the first transferable image attacks on closed-source large vision language models which can be viewed as early, simple versions of jailbreaks.

We believe that our findings not only advance the field of adversarial robustness but also provide valuable insights into the nature of neural network representations and their vulnerability to adversarial perturbations. The connection between interpretability and robustness highlighted in this work also opens up new research directions for developing more reliable and explainable AI systems.

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

# A  ADDITIONAL INSIGHTS AND APPLICATIONS

We want to support our multi-resolution input choice as an active defense by demonstrating that by reversing it and representing an adversarial perturbation *explicitly* as a sum of perturbations at different resolutions, we get human-interpretable perturbations by default.

## A.1  SINGLE-RESOLUTION ADVERSARIAL ATTACKS

Natural images contain information expressed on all frequencies, with an empirically observed power-law scaling. The higher the frequency, the lower the spectral power, as $\propto f^{-2}$ (van der Schaaf & van Hateren, 1996).

While having a single perturbation $P$ of the full resolution $R \times R$ theoretically suffices to express anything, we find that this choice induces a specific kind of high frequency prior. Even simple neural networks can theoretically express any function (Hornik et al., 1989), yet the specific architecture matters for what kind of a solution we obtain given our data, optimization, and other practical choices. Similarly, we find that an alternative formulation of the perturbation $P$ leads to more natural looking and human interpretable perturbations despite the attacker having access to the highest-resolution perturbation as well and could in principle just use that.

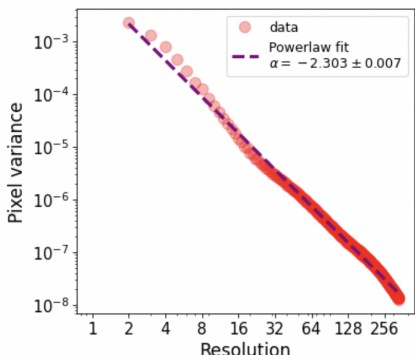

Figure 11: The image spectrum of generated multi-resolution attacks. The adversarial attacks generated over multiple resolutions at once end up showing very white-noise-like distribution of powers over frequencies (the slope for natural images is $\approx -2$). This is in contrast with standard noise-like attacks.

## A.2  MULTI-RESOLUTION ATTACKS

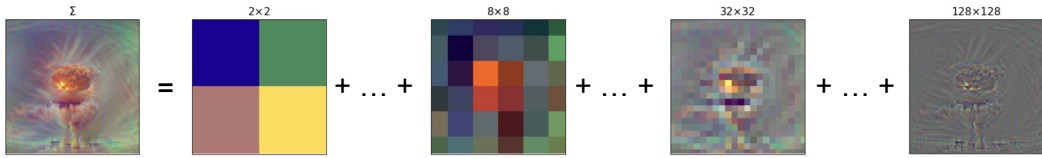

Figure 12: The result of expressing an image as a set of resolutions and optimizing it towards the CLIP embedding of the text '*a photo of a nuclear explosion*'. The plot shows the resulting sum of resolutions (left panel, marked with $\rho$) and selected individual perturbations $P_r$ of resolutions $2 \times 2$, $8 \times 8$, $32 \times 32$ and $128 \times 128$. The intensity of each is shifted and rescaled to fit between 0 and 1 to be recognizable visually, however, the pixel values in the real $P_r$ fall of approximately as $r^{-1}$.

We express the single, high resolution perturbation $P$ as a sum of perturbations $P = \sum_{r \in \rho} \text{rescale}_R(P_r)$, where $P_r$ is of the resolution $r \times r$ specified by a set of resolutions $\rho$, and the $\text{rescale}_R$ function rescales and interpolates an image to the full resolution $R \times R$. When we jointly optimize the set of perturbations $\{P_r\}_{r \in \rho}$, we find that: a) the resulting attacked image $X + \sum_{r \in \rho} \text{rescale}_R(P_r)$ is much more human-interpretable, b) the attack follows a power distribution of natural images.

When attacking a classifier, we choose a target label $t$ and optimize the cross-entropy loss of the predictions stemming from the perturbed image as if that class $t$ were ground truth. To add to the robustness and therefore interpretability of the attack (as hypothesized in our *Interpretability-Robustness Hypothesis*), we add random jitter in the x-y plane and random pixel noise, and design the attack to work on a set of models.

An example of the multi-resolution sum is show in Figure 13. There we use a simple Stochastic Gradient Descent (Robbins & Monro, 1951) optimization with the learning rate of $5 \times 10^{-3}$ and a cosine decay schedule over 50 steps. We add a random pixel noise of $0.6$ (out of 1), jitter in the x-y plane in the $\pm 5$ range and a set of all perturbations from $1 \times 1$ to $224 \times 224$ interpolated using `bicubic` interpolation (Keys, 1981). In Figure 13 we see that despite the very limited expressiveness

of the final layer class label, we can still recover images that look like the target class to a human. We also tested them using Gemini Advanced and GPT-4, asking what the AI model sees in the picture, and got the right response in all 8 cases. To demonstrate that we can generate images beyond the

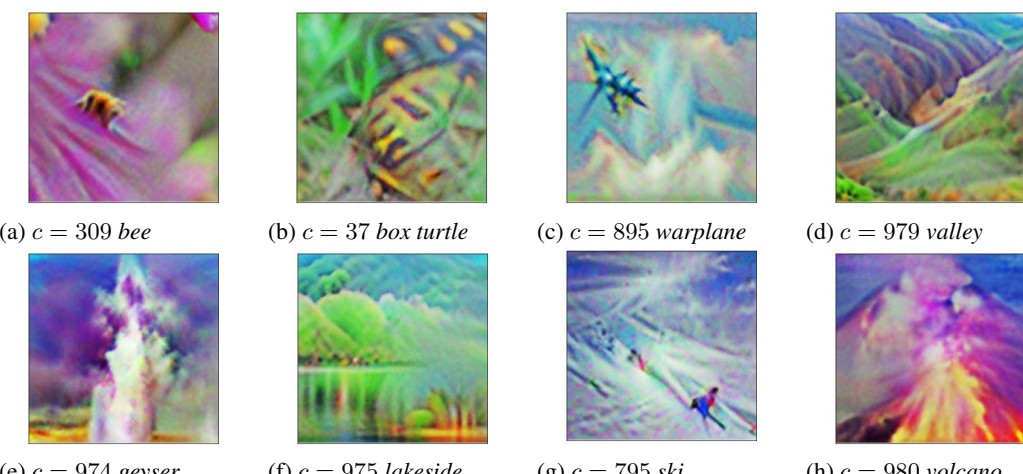

(a) $c = 309$ *bee*   (b) $c = 37$ *box turtle*   (c) $c = 895$ *warplane*   (d) $c = 979$ *valley*

(e) $c = 974$ *geyser*   (f) $c = 975$ *lakeside*   (g) $c = 795$ *ski*   (h) $c = 980$ *volcano*

Figure 13: Examples of images generated as attacks on ImageNet-trained classifiers. These images were generated by minimizing the cross-entropy loss of seven pretrained classifiers with respect to the target ImageNet class. Spatial jitter in the $\pm 5$ pixel range and pixel noise of standard deviation 0.6 were applied during SGD optimization with learning rate $5 \times 10^{-3}$ over 50 steps with a cosine schedule. The perturbation was expressed as a sum of perturbations at all resolutions from $1 \times 1$ to $224 \times 224$ that were optimized at once.

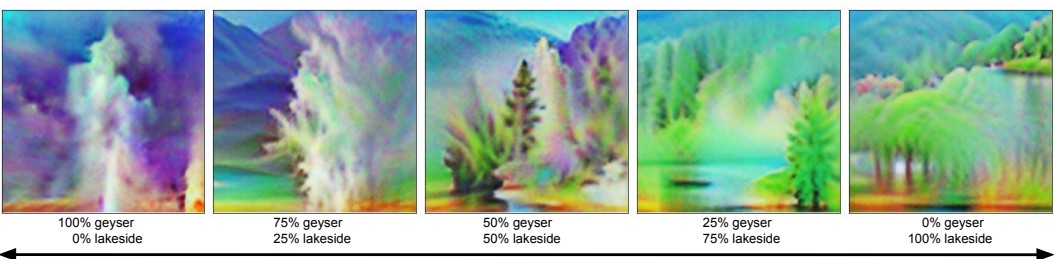

Figure 14: Optimizing towards a probability vector with a sliding scale between $c = 974$ *geyser* and $c = 975$ *lakeside*. Optimizing against pretrained classifiers generated semantically blended image of the two concepts.

original 1000 ImageNet classes, we experimented with setting the target label not as a one-hot vector, but rather with target probability $p$ on class $t_1$ and $1 - p$ on $t_2$. For classes $c = 974$ (*geyser*) and $c = 975$ (*lakeside*) we show, in Figure 14 that we get semantically meaningful combinations of the two concepts in the same image as we vary $p$ from 0 to 1. $p = 1/2$ gives us a *geyser* hiding beyond trees at a *lakeside*. This example demonstrates that in a limited way, classifiers can be used as controllable image generators.

## A.3 MULTI-RESOLUTION ATTACK ON CLIP

The CLIP-style (Radford et al., 2021) models map an image $I$ to an embedding vector $f_I : I \to v_I$ and a text $T$ to an embedding vector $f_T : T \to v_T$. The cosine between these two vectors corresponds to the semantic similarity of the image and the text, $\cos(v_I, v_T) = v_I \cdot v_T / (|v_I| |v_T|)$. This gives us $\text{score}(I, T)$ that we can optimize.

Adversarial attacks on CLIP can be thought of as starting with a human-understandable image $X_0$ (or just a noise), and a target label text $T^*$, and optimizing for a perturbation $P$ to the image that tries to

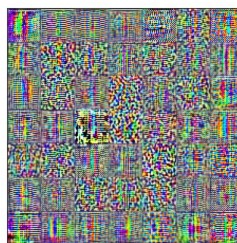
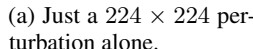
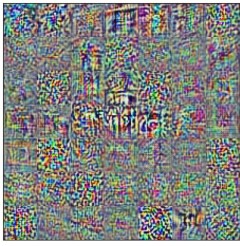
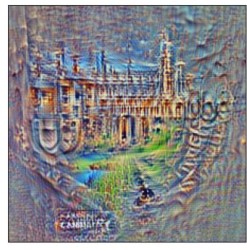
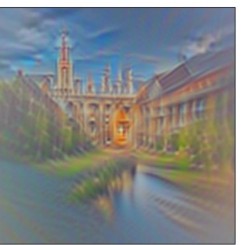

(a) Just a $224 \times 224$ perturbation alone.
(b) Adding random noise to optimization.
(c) Adding random jitter to optimization.
(d) Adding all resolutions from $1 \times 1$ to $224 \times 224$.

Figure 15: The effect of adding noise, jitter, and a full set of resolutions to an adversarial attack on CLIP towards the text *'a beautiful photo of the University of Cambridge, detailed'*. While using just a plain perturbation of the full resolution in Figure 15a, as is standard in the typical adversarial attack setup, we get a completely noise-like image. Adding random noise to the pixels during optimization leads to a glimpse of a structure, but still maintains a very noise-like pattern (Figure 15b). Adding random jitter in the x-y plane on top, we can already see interpretable shapes of *Cambridge* buildings in Figure 15c. Finally, adding perturbations of all resolutions, $1 \times 1, 2 \times 2, \ldots, 224 \times 224$, we get a completely interpretable image as a result in Figure 15d.

increase the $\text{score}(X_0 + P, T^*)$ as much as possible. In general, finding such perturbations is easy, however, they end up looking very noise-like and non-interpretable. (Fort, 2021b;a).

If we again express $P = \text{rescale}_{224}(P_1) + \text{rescale}_{224}(P_2) + \cdots + P_{224}$, where $P_r$ is a resolution $r \times r$ image perturbation, and optimize $\text{score}(X_0 + \text{rescale}_{224}(P_1) + \text{rescale}_{224}(P_2) + \cdots + P_{224}, T^*)$ by simultaneously updating all $\{P_r\}_r$, the resulting image $X_0 + \sum_{r \in [1,224]} \text{rescale}_R(P_r)$ looks like the target text $T^*$ to a human rather than being just a noisy pattern. Even though the optimizer could choose to act only on the full resolution perturbation $P_{224}$, it ends up optimizing all of them jointly instead, leading to a more natural looking image. To further help with natural-looking attacks, we introduce pixel noise and the x-y plane jitter, the effect of which is shown in Figure 15.

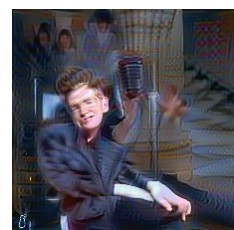

Figure 16: An attack on vision language models. GPT-4 sees *Rick Astley from his famous "Never Gonna Give You Up" music video* tree. See Table 21 and 22 for details.

We use SGD at the learning rate of $5 \times 10^{-3}$ for 300 steps with a cosine decay schedule to maximize the cosine between the text description and our perturbed image. We use the `OpenCLIP` models (Ilharco et al., 2021; Cherti et al., 2023) (an open-source replication of the CLIP model (Radford et al., 2021)). Examples of the resulting "adversarial attacks", starting with a blank image with $0.5$ in its RGB channels, and optimizing towards the embedding of specific texts such as *"a photo of Cambridge UK, detailed*, and *"a photo of a sailing boat on a rough sea"* are shown in Figure 18. The image spectra are shown in Figure 11, displaying a very natural-image-like distribution of powers. The resulting images look very human-interpretable.

Starting from a painting of Isaac Newton and optimizing towards the embeddings of *"Albert Einstein"*, *"Queen Elizabeth"* and *"Nikola Tesla"*, we show that the attack is very semantically targeted, effectively just changing the facial features of Isaac Newton towards the desired person. This is shown in Figure 17. This is exactly what we would ideally like adversarial attacks to be – when changing the content of what the model sees, the same change should apply to a human. We use a similar method to craft transferable attacks (see Figure 16 for an example) against commercial, closed source vision language models (GPT-4, Gemini Advanced, Claude 3 and Bing AI) in Table 21, in which a *turtle* turns into a *cannon*, and in Table 22, where *Stephen Hawking* turns into the music video *Never Gonna Give You Up* by *Rick Astley*. The attacks also transfer to Google Lens, demonstrating that the multi-resolution prior also serves as a good *transfer* prior and forms an early version of a transferable image vision language model jailbreak. This is a constructive proof to the contrary of the non-transferability results in Schaeffer et al. (2024).

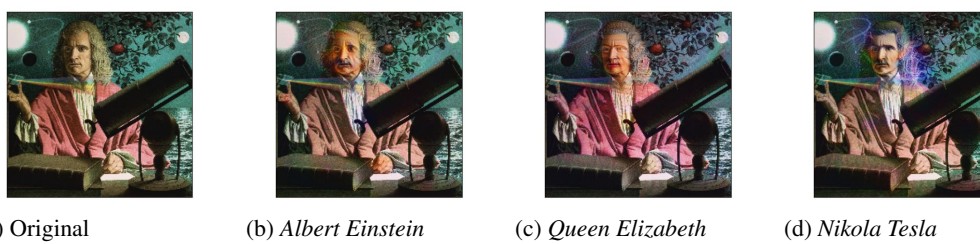

(a) Original      (b) *Albert Einstein*      (c) *Queen Elizabeth*      (d) *Nikola Tesla*

Figure 17: Starting with an image of Isaac Newton and optimizing a multi-resolution perturbation towards text embeddings of *Albert Einstein*, *Queen Elizabeth* and *Nikola Tesla* leads to a change in the face of the person depicted. This demonstrates how semantically well-targeted such multi-resolution attacks are. All 4 images are recognizable as the target person to humans as well as GPT-4o and Gemini Advanced.

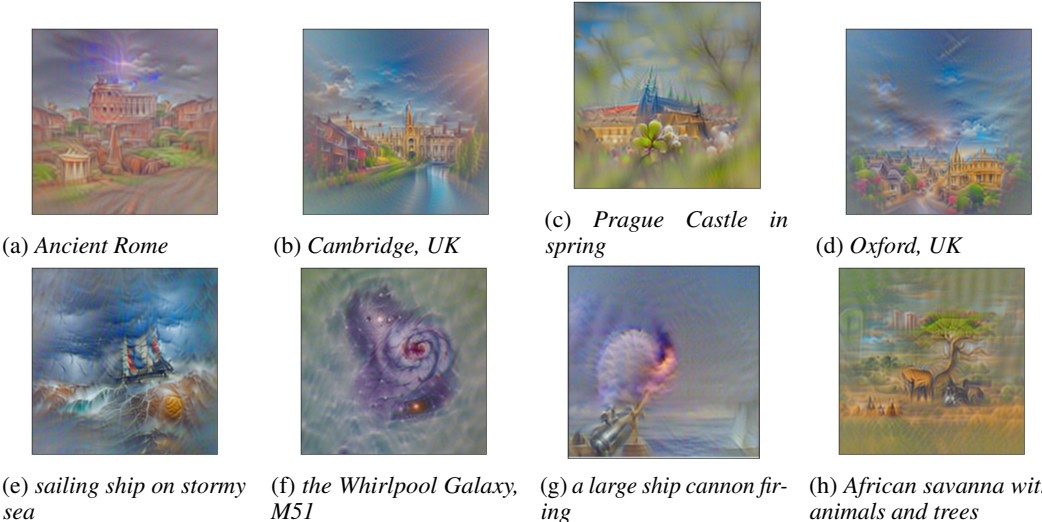

(a) *Ancient Rome*    (b) *Cambridge, UK*    (c) *Prague Castle in spring*    (d) *Oxford, UK*

(e) *sailing ship on stormy sea*    (f) *the Whirlpool Galaxy, M51*    (g) *a large ship cannon firing*    (h) *African savanna with animals and trees*

Figure 18: Examples of images generated with the multi-resolution prior, jitter and noise with the OpenCLIP models. The text whose embedding the image optimizes to approach is of the form 'A beautiful photo of [X], detailed' for different values of [X].

## A.4 CROSSMAX EXPERIMENTS

To demonstrate experimentally different characteristics of prediction aggregation among several classifiers, we trained 10 ResNet18 models, starting from an ImageNet pretrained model, changing their final linear layer to output 10 classes of CIFAR-10. We then used the first 2 attacks of the RobustBench `AutoAttack` suite (APGD-T and APGD-CE; introduced by Croce & Hein (2020) as particularly strong attack methods) and evaluated the robustness of our ensemble of 10 models under adversarial attacks of different $L_\infty$ strength. The results are shown in Figure 19.

The aggregation methods we show are 1) our CrossMax (Algorithm 1) (using *median* since the 10 models are expected to be equally good), 2) a standard logit mean over models, 3) median over models, and 4) the performance of the individual models themselves. While an ensemble of 10 models, either aggregated with a mean or median, is more robust than individual models at all attack strengths, it nonetheless loses robust accuracy very fast with the attack strength $L_\infty$ and at the standard level of $L_\infty = 8/255$ it drops to ≈0%. Our *CrossMax* in Algorithm 1 provides $> 0$ robust accuracy even to $10/255$ attack strengths, and for $8/255$ gives a 17-fold higher robust accuracy than just plain mean or median.

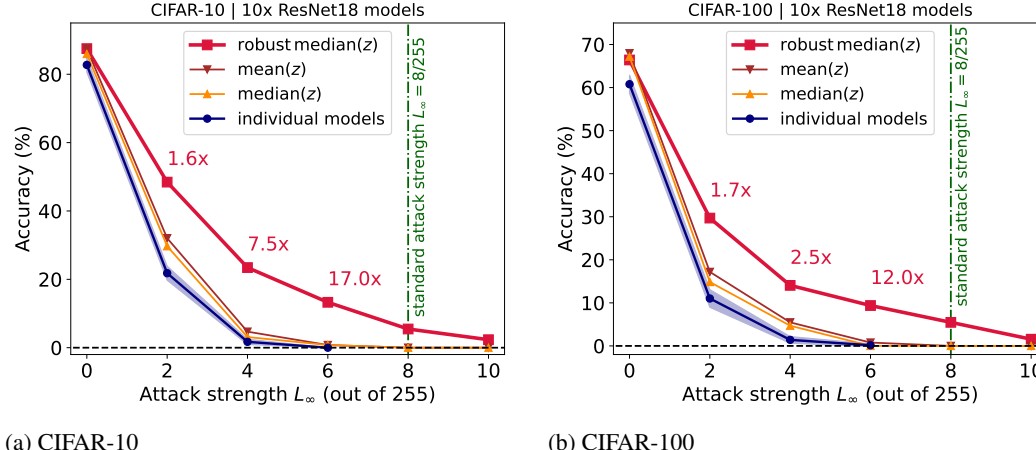

(a) CIFAR-10         (b) CIFAR-100

Figure 19: The robust accuracy of different types of ensembles of 10 ResNet18 models under increasing $L_\infty$ attack strength. Our robust median ensemble, *CrossMax*, gives very non-trivial adversarial accuracy gains to ensembles of individually brittle models. For $L_\infty = 6/255$, its CIFAR-10 robust accuracy is 17-fold larger than standard ensembling, and for CIFAR-100 the factor is 12.

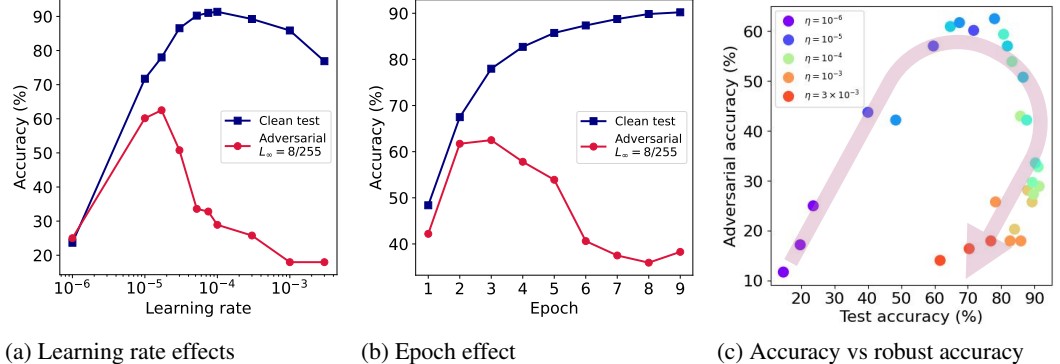

(a) Learning rate effects     (b) Epoch effect     (c) Accuracy vs robust accuracy

Figure 20: Finetuning a pretrained model with multi-resolution inputs. The left panel shows the test accuracy and adversarial accuracy after the first two attacks of RobustBench AutoAttack at $L_\infty = 8/255$ after 3 epochs of finetuning an ImageNet pretrained ResNet152. The middle panel shows the effect of training epoch for a single finetuning run at the learning rate $\eta = 1.7 \times 10^{-5}$. The right panel shows a hysteresis-like curve where high test accuracies are both compatible with low and high adversarial accuracies. The test accuracies are over the full 10,000 images while the adversarial accuracies are evaluated on 128 test images.

## A.5   FINETUNING EFFECTS

## A.6   DETAILS OF ADVERSARIAL FINETUNING

## A.7   TRANSFER TO MASSIVE COMMERCIAL MODELS

In Table 21 we show the results of asking *"What do you see in this photo?"* and adding the relevant picture to four different, publicly available commercial AI models: GPT-4[1], Bing Copilot[2], Claude 3 Opus[3] and Gemini Advanced[4]. We find that, with an exception of Gemini Advanced, even a

---

[1] chatgpt.com
[2] bing.com/chat
[3] claude.ai/
[4] gemini.google.com

| Dataset | Adv. train | Model | Method | # | Test acc | rand RobustBench AutoAttack $L_\infty = 8/255$ # samples (%) | | |
|---|---|---|---|---|---|---|---|---|
| | | | | | | Adv acc | APGD→ CE | APGD DLR |
| CIFAR-10 | ✓ | ResNet152 | Multi-res backbone | 128 | 87.19 | 46.88 | 34.38 | 32.03 |
| CIFAR-10 | ✓ | ResNet152 | Self-ensemble | 128 | 84.58 | 67.94 | 64.06 | 54.69 |
| CIFAR-10 | ✓ | ResNet152 | 3-ensemble of self-ensembles | 128 | 87.00 | **78.13** | 73.44 | 72.65 |
| CIFAR-10 | ✓ | [3] | SOTA #1 | | | 73.71 | | |
| CIFAR-100 | ✓ | ResNet152 | Multi-res backbone | 128 | 62.72 | 37.50 | 32.03 | 22.66 |
| CIFAR-100 | ✓ | ResNet152 | Self-ensemble | 512 | 58.93 | **47.85** ±2.66 | 36.72 ±3.01 | 33.98 ±2.72 |
| CIFAR-100 | ✓ | ResNet152 | 3-ensemble of self-ensembles | 512 | 61.17 | **51.28** ±1.95 | 44.60 ±2.00 | 43.04 ±1.97 |
| CIFAR-100 | ✓ | [48] | SOTA #1 | | | 42.67 | | |

Table 2: Full *randomized* (=the strongest against our approach) RobustBench AutoAttack adversarial attack suite results for 128 test samples at the $L_\infty = 8/255$ strength. In this table we show the results of attacking our multi-resolution ResNet152 models finetuned on CIFAR-10 and CIFAR-100 from an ImageNet pretrained state **with** light adversarial training.

$L_\infty = 30/255$ attack generated in approximately 1 minute on a single A100 GPU (implying a cost at most in cents) fools these large models into seeing a *cannon* instead of a *turtle*. The attack also transfers to Google Lens.

## A.8 ATTACK TRANSFER BETWEEN LAYERS

# B VISUALIZING ATTACKS ON MULTI-RESOLUTION MODELS

# C ADDITIONAL EXPERIMENTS FOR CROSSMAX

# D ADDITIONAL CROSSMAX VALIDATION

As an ablation, we tested variants of the *CrossMax* method. There are two normalization steps: A) subtracting the per-predictor max, and B) subtracting the per-class max. We exhaustively experiment with all combinations, meaning $\{\_, A, B, AB, BA\}$, (robust accuracies at 4/255 are $\{4, 4, 0, 22, 0\}\%$) and find that performing $A$ and then $B$, as in Algorithm 1, is by far the most robust method. We perform a similar ablation for a robust, multi-resolution self-ensemble model in Table 3 and reach the same verdict, in addition to confirming that the algorithm is very likely not accidentally masking gradients.

## D.1 TRAINING FROM SCRATCH

For our ResNet18 model trained from scratch on CIFAR-10, we keep the pairs of images that are mixed in `mixup` fixed for 20 epochs at a time, producing a characteristic pattern in the training accuracies. Every 5 epochs we re-draw the random mixing proportions in the $[0, 1/2]$ range. We trained the model for 380 epochs with the Adam optimizer (Kingma & Ba, 2015) at learning rate $10^{-3}$ and dropped it to $10^{-4}$ for another 120 epochs. The final checkpoint is the weight average of the last 3 epochs. The training batch size is 512. These choices are arbitrary and we did not run a hyperparameter search over them.

Figure 21: Multi-resolution adversarial attacks of increasing $L_\infty$ using OpenCLIP on an image of a sea turtle towards the text *"a cannon"* tested on GPT-4, Bing Copilot (Balanced), Claude 3 Sonnet and Gemini Advanced. All models we tested the images on were publicly available. The conversation included a single message *"What do you see in this photo?"* and an image. We chose the most relevant parts of the response.

| | Original | $L_\infty = 20/255$ | $L_\infty = 30/255$ | $L_\infty = 40/255$ | $L_\infty = 70/255$ | $L_\infty = 100/255$ |
|---|---|---|---|---|---|---|
| |  |  |  |  |  |  |
| GPT-4 | sea turtle swimming | turtle swimming in water | **cannon**, mounted on stone base, firing | **cannon** with a notably ornate and rusted appearance | **cannon** mounted on a brick platform | stylized or artistically rendered depiction of a **cannon** |
| Bing Copilot | sea turtle gracefully swimming | sea turtle gracefully swimming | a **cannon** mounted on a stone base | **cannon** with a wheel, mounted on a stone base | old **cannon** mounted on a brick platform | color-saturated **cannon** mounted on wheels |
| Claude 3 Opus | sea turtle swimming in clear, turquoise water | sea turtle swimming underwater | old **cannon** submerged underwater | old decorative **cannon** sitting on a stone or concrete platform | old naval **cannon** set on a stone or brick platform | artistic painting or illustration of an old **cannon** |
| Gemini Advanced | sea turtle swimming underwater | sea turtle swimming underwater | sea turtle swimming | sea turtle swimming in a pool | **cannon** being fired by a **turtle** wearing a red jacket | artistic interpretation of a **cannon** firing |

| Aggregation fn | $\text{topk}_2$ | | | | | mean | | | | |
|---|---|---|---|---|---|---|---|---|---|---|
| Method | _ | A | B | BA | AB | _ | A | B | BA | AB |
| Test acc | 57.08 | 59.86 | 0.82 | 1.27 | 58.92 | 60.31 | 59.89 | 1.1 | 1.05 | **57.23** |
| Adv acc | 46.88 | 46.88 | 1.56 | 0.00 | 57.81 | 40.62 | 48.44 | 0.00 | 0.00 | 39.06 |

Table 3: CrossMax algorithm ablation. The Algorithm 1 contains two subtraction steps: A = the per-predictor max subtraction, and B = the per-class max subtraction. This Table shows the robust accuracies of a self-ensemble model on CIFAR-100 trained with light adversarial training, whose intermediate layer predictions were aggregated using different combinations and orders of the two steps. We also look at the effect of using the final $\text{topk}_2$ aggregation vs just using a standard mean. The best result is obtained by the Algorithm 1, however, we see that not using the topk does not lead to a critical loss of robustness as might be expected if there were accidental gradient masking happening.

Figure 22: Multi-resolution adversarial attacks of increasing $L_\infty$ using OpenCLIP on an image of *Stephen Hawking* towards the embedding of an image from the famous *Rick Astley's* song *Never Gonna Give You Up* from the 1980s tested on GPT-4, Bing Copilot (Balanced), Claude 3 Sonnet and Gemini Advanced. All models we tested the images on were publicly available. The conversation included a single message *"What do you see in this photo?"* and an image. We chose the most relevant part of the response. Unfortunately, Gemini refused to answer, likely due to the presence of a human face in the photo.

| | Original | $L_\infty = 20/255$ | $L_\infty = 30/255$ | $L_\infty = 40/255$ | $L_\infty = 70/255$ | $L_\infty = 100/255$ |
|---|---|---|---|---|---|---|
| |  |  |  |  |  |  |
| GPT-4 | Stephen Hawking | Stephen Hawking | Never Gonna Give You Up | Never Gonna Give You Up | Never Gonna Give You Up | singer or performer, possibly Rick Astley |
| Bing Copilot | individual sitting in a wheelchair | individual sitting on a bench | individual sitting down, holding a microphone, singing | person seated, holding a musical instrument | two individuals in an indoor setting | person in front of a microphone, singing |
| Claude 3 Opus | elderly man in a wheelchair | man in a wheelchair, smiling | young man with blonde hair, vintage-style microphone, singing | young man with blond hair, 1980s pop music | music video, 1980s, singer | music video, 1980s fashion |
| Gemini Advanced | Refused to answer. | Refused to answer. | Refused to answer. | Refused to answer. | Refused to answer. | Refused to answer. |

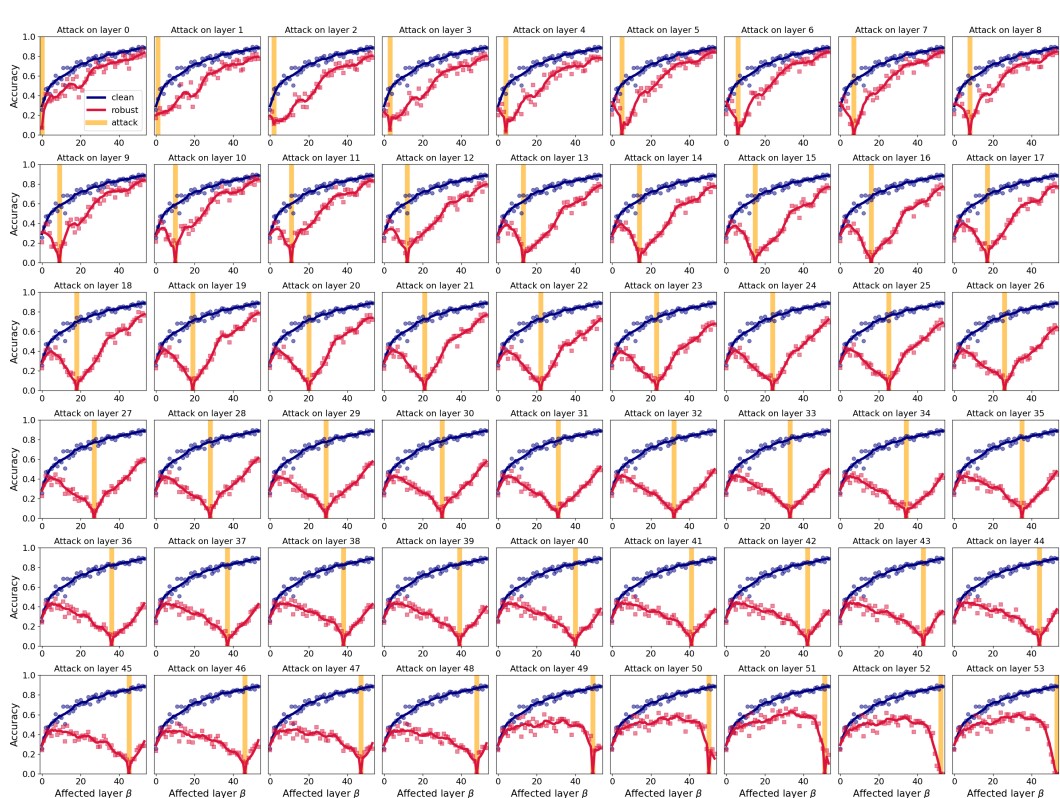

Figure 23: Attack transfer between layers of the ResNet154 model pre-trained on ImageNet-1k. The individual linear heads were finetuned on CIFAR-10 on top of the frozen model.

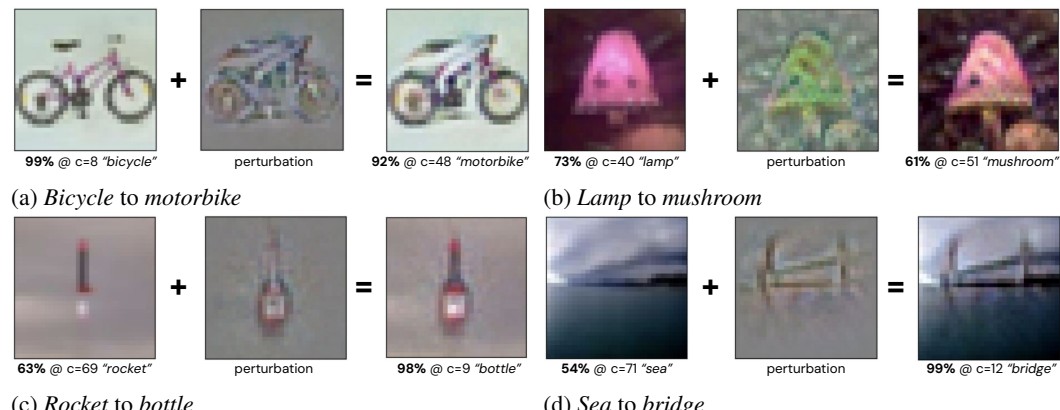

(a) *Bicycle* to *motorbike*

(b) *Lamp* to *mushroom*

(c) *Rocket* to *bottle*

(d) *Sea* to *bridge*

Figure 24: Additional examples of an adversarial attack on an image towards a target label. We use simple gradient steps with respect to our multi-resolution ResNet152 finetuned on CIFAR-100. The resulting attacks use the underlying features of the original image and make semantically meaningful, human-interpretable changes to it. Additional examples available in Figure 8.

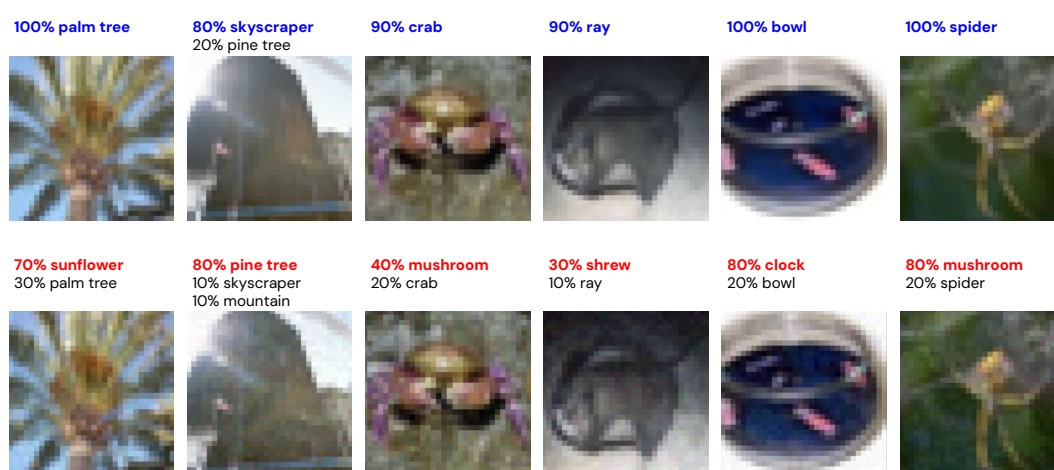

Figure 25: Examples of successfully attacked CIFAR-100 images for an ensemble of self-ensembles – our most robust model. We can see human-plausible ways in which the attack changes the perceived class. For example, the skyscraper has a texture added to it to make it look tree-like.

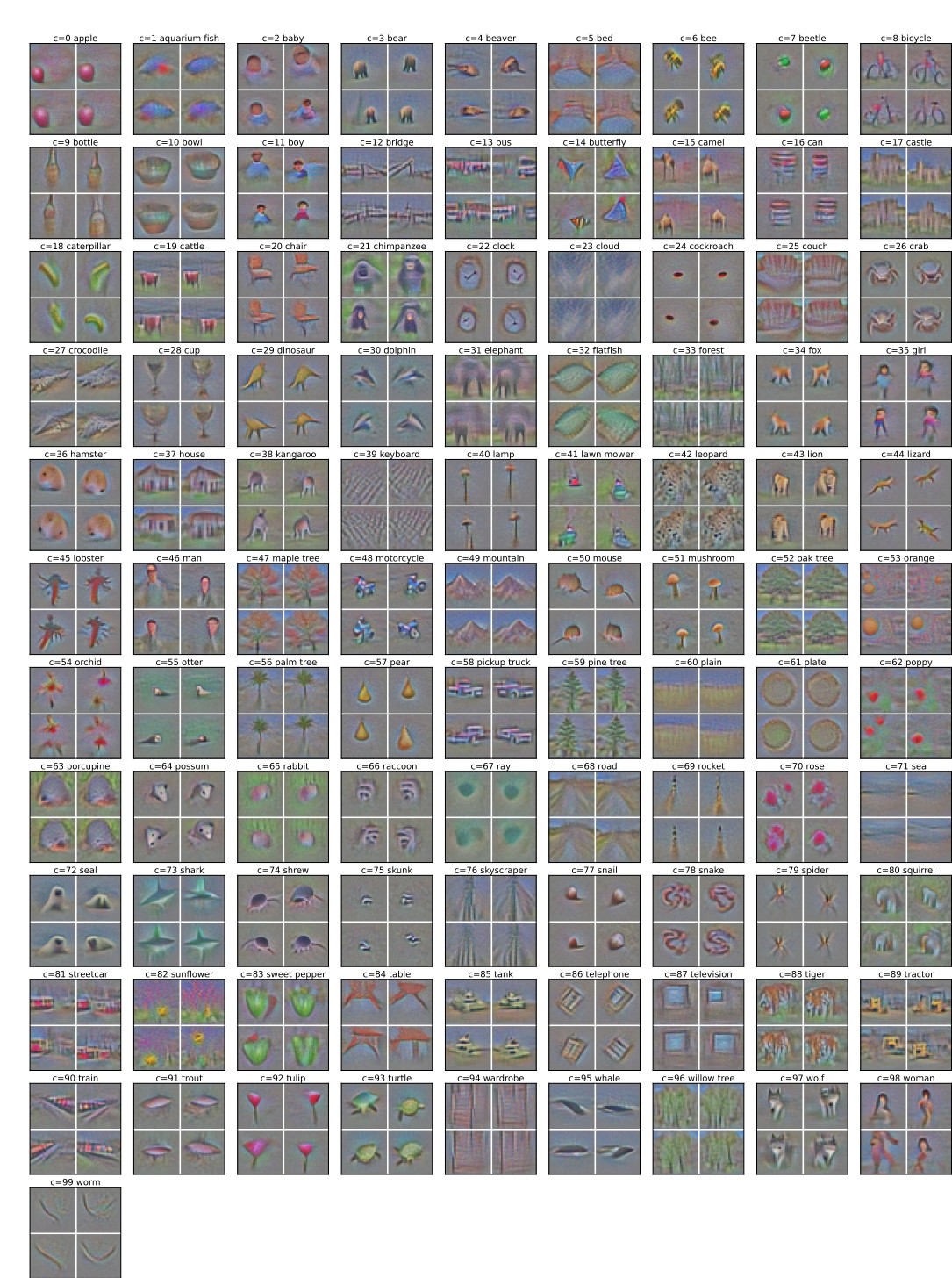

Figure 26: Examples of optimizing towards all 100 CIFAR-100 classes against our multi-resolution ResNet152 model, 4 examples for each. We use 400 simple gradient steps at learning rate $\eta = 1$ with SGD with respect to the model, starting from all grey pixels (128,128,128). The resulting attacks are easily recognizable as the target class to a human.

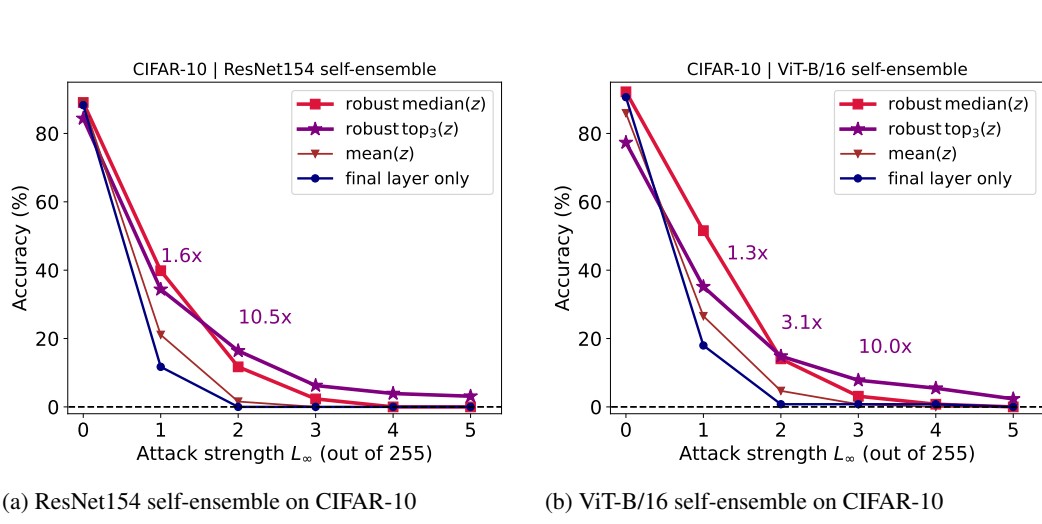

(a) ResNet154 self-ensemble on CIFAR-10        (b) ViT-B/16 self-ensemble on CIFAR-10

Figure 27: The robust accuracy of different types of self-ensembles of ResNet152 and ViT-B/16 with linear heads finetuned on CIFAR-10 under increasing $L_\infty$ attack strength.

