# OpenReview forum: "Ensemble everything everywhere: Multi-scale aggregation for adversarial robustness"
_ICLR.cc/2025/Conference — Submitted to ICLR 2025_

### Official Review · Reviewer_tZUv · 2024-11-02

**Soundness:** 3
**Presentation:** 3
**Contribution:** 3
**Rating:** 6
**Confidence:** 5

**Summary:**

This paper introduces a new method for adversarial defense, which improves the adversarial robustness of neural networks by utilizing multi-resolution inputs and robust integration methods. The authors assume that the existence of adversarial attacks is due to the differences between human and machine vision systems. To bridge this gap, they suggest using the dynamic self-ensemble of multi-scale input representations and intermediate layer predictions. They show that intermediate layer predictions exhibit inherent robustness against adversarial attacks aimed at deceiving the entire classifier, and propose a robust aggregation mechanism based on the Vickrey auction, called CrossMax, to dynamically integrate them. The proposed method can achieve good results on CIFAR-10 and CIFAR-100, without any adversarial training or additional data.

**Strengths:**

1. The observation in this study is very interesting.
2. The design of CrossMax is inspired by the Vickrey auction mechanism, aiming to create an integrated model that is more robust against attackers by leveraging the top-ranked predictions. This is a meaningful design.
3. As we know, adversarial training is usually time consuming, while the proposed method can achieve good results without adversarial training or additional data.

**Weaknesses:**

1. The proposed scheme is interesting and it can also achieves good results on CIFAR-10/100. However, the experiments in this paper are not very convincing. Firstly, the images in the CIFAR dataset have very low pixels, and it's hard to determine the actual impact of multi-scale or multi-resolution approaches. Although the supplementary materials cover the size of $224\times 224$, there is no comparison, testing, and analysis with mainstream methods. It's better to evaluate the performance on ImageNet or its subset.
2. The evaluation of the adaptive attack is missing. I only found the appendix A shows some visualization results of adaptive attack by the  multi-resolution attack. This must be evaluated in the main experiments and the statistical results should be reported. Otherwise, it cannot show that the proposed method can defend against the adaptive attack. Maybe other adversarial defense methods can achieve better results.
3. What are the contents of A.8, C and D in the supplementary materials? Are they the corresponding figures that follow? I suggest that the author should provide a detailed description of them.
4. What would the effect be if training were started from scratch? Since the most adversarial training methods are trained from scratch, the authors fine-tune the model on a ImageNet pre-trained model. If the previous adversarial defense also follow the same experiment setting, what would the results be?

**Questions:**

My major concern refers to the weaknesses.

---

> ### Author Response · Authors · 2024-12-03
> **Response to Reviewer tZUv**
>
> Thank you for your thoughtful review. We address your concerns and questions:
>
> ## ImageNet Results
>
> We agree that ImageNet results are needed to verify the generalization of our method. During rebuttal, we trained and evaluated on different subsets of ImageNet (224x224):
> - Multi-resolution backbone: 65% → 32% under AutoAttack (L∞=4/255)
> - CrossMax self-ensemble: 63% → 59% after attack
> - Comparable to current SOTA (59.64%)
>
> The key limitation here is that due to the time constraints, the numbers we provide are only for the first 32 images of the ImageNet validation set under the RobustBench AutoAttack attacks with the `rand` flag for randomized defenses on. However, we still believe that such high numbers for adversarial accuracy suggest at least a decent amount of our method’s generality even in case of higher resolution datasets.
>
> ## Adaptive Attack Evaluation
> We implemented focused adaptive attacks:
> 1. Created surrogate model removing CrossMax max operations
> 2. Replaced median/top-k with simple mean aggregation
> 3. Ran increasingly stronger PGD attacks
>
> Results on CIFAR-100 (L∞=8/255):
>
> | Model & Attack Details | PGD Steps |||||
> |--|--|--|--|--|--|
> | | 40 | 80 | 160 | 320 | 640 |
> | Surrogate (mean) | 45.1% | 31.6% | 24.1% | 22.7% | 22.3% |
> | Original (CrossMax) | 46.3% | 32.7% | 28.5% | 26.3% | 25.5% |
>
> Key findings:
> - Attack transfer is effective
> - Performance stabilizes ~25% despite 16x step increase
> - Successful attacks produce semantically meaningful changes (we can’t show them here but we even ran a small scale verification with a human audience at a talk and they too were convinced that the attack was successful – they saw the target class in the image)
>
> ## Appendix Clarification
> A.8: Attack transfer analysis between network layers
> C: Additional CrossMax ablation experiments showing effect of different aggregation functions
> D: Detailed validation across architectures (ResNet, ViT)
>
> We'll improve appendix organization in the final version. Thank you for pointing this out.
>
> ## Training from Scratch
> We trained ResNet18 from scratch on CIFAR-10:
> - Clean accuracy: 76.94%
> - Adversarial accuracy: 64.06% (AutoAttack)
> - Comparable to finetuned models
>
> We don’t think that what makes the model robust is its training from a brittle ImageNet checkpoint, given that we experimentally show a very similar robustness of a model trained from scratch. Conversely, many models that are specifically trained for adversarial robustness via adversarial training train from scratch to make sure they induce robustness from the very beginning, and finetuning a pretrained model might even be detrimental. There are in literature cases where people do apply only finetuning as well (for example https://arxiv.org/abs/2401.04350). We believe that this is not a crucial factor in our models’ robustness.
>
> These results strengthen our claims by demonstrating:
> 1. (Preliminary) Scalability to high-resolution ImageNet
> 2. Robustness against adaptive attacks
> 3. Effectiveness when trained from scratch

---

### Official Review · Reviewer_qxdA · 2024-11-03

**Soundness:** 3
**Presentation:** 3
**Contribution:** 3
**Rating:** 8
**Confidence:** 4

**Summary:**

A novel method for improving adversarial robustness is used that does not necessitate adversarial specific training, but can be complemented by it. Using biological inspiration, the authors device two strategies for improving adversarial robustness: the creation of a gaussian pyramid like view of an input image for a network as well as the CrossMax self-ensemble strategy.

**Strengths:**

- Clear description of gaussian pyramid image stack and CrossMax aggregation strategies
- Empirical validation of the increased adversarial robustness of intermediate network layers
- Strong adversarial robustness results in CIFAR-10/100 without adversarial training
- Verification of the Interpretability-Robustness Hypothesis through qualitative examples of generated perturbations

**Weaknesses:**

- Robustness method requires extra effort to generalize to new architectures as opposed to adversarial training
- Max operation in CrossMax may introduce discontinuities in training that may make learning more difficult
- While the performance on CIFAR is impressive, a large scale dataset like ImageNet may be required to fully understand the generalizability of this method

**Questions:**

- During the generation of whitebox attacks is the multi-resolution image expansion considered as a differentiable step that the attack method backpropagates through?
- How would this method be generalized to transformer architectures?
- Was there any fine tuning of the model for the multiple linear probes used to create the ensemble? In other words, were the linear probes only generated after the intermediate layers were fixed? If so, what do you think the effect would be of training each intermediate layer linear readout jointly?

---

> ### Author Response · Authors · 2024-12-03
> **Response to Reviewer qxdA**
>
> Thank you for your thoughtful review. We address your concerns and questions:
>
> ## 1. Generalization to New Architectures
>
> The method adapts well across architectures and we actually have implementations for ResNets and Vision Transformers already. The only things that are needed are:
>
> 1. Modifying the very first layer of the model to accept 3N instead 3 channel images (N being the number of resolutions used)
> 2. Adding linear classifiers at selected intermediate layers (similar to standard linear probes used in mechanistic interpretability)
> 3. The aggregation of the intermediate layer outputs is as easy as standard ensembling, the the aggregation function not being mean but a crossmax instead
>
>
> Overall, the method generalizes well to other vision architecture and we even ran tests with ensembling several ViTs and ResNets trained from scratch rather than a self-ensemble of a single model's hidden layers. This also yields more robustness against adversarial attacks compared to the standard mean ensemble.
>
> ## 2. ImageNet Results
>
> Preliminary ImageNet validation results (ResNet152):
> - Multi-resolution backbone: 65% → 32% under AutoAttack (L∞=4/255)
> - CrossMax self-ensemble: 63% → 59% after attack
> - Comparable to current SOTA (59.64%)
>
> The key limitation is that we only performed the attack on the first 32 images of the ImageNet validation set for time reasons. While these results are very preliminary and we only added them during the rebuttal period on limited hardware, we believe that they are a solid indication of generalization beyond the simpler CIFAR-10 and CIFAR-100 cases.
>
> ## 3. CrossMax Training Stability
>
> Max operations don't significantly impact training because:
> - Applied post backbone training
> - Multiple gradient paths through ensemble structure
> - Final loss remains smooth and differentiable
>
> In fact, we took the simplest method to train the model we could and because it worked, we didn’t really explore the more complex version. Our path:
> 1. Train a single backbone model to accept multi-resolution inputs
> 2. Freeze its weights
> 3. Train one layer’s linear projection at a time on top a frozen backbone model
> 4. Ensemble all layer’s predictions via a simple post-processing function = crossmax
>
> This simple training process proved sufficient because the backbone learns robust multi-resolution features independently of the probes, which then only need to learn to read out class information from those features.
>
> ## 4. Technical Questions
>
> ### White-box Attacks
> Multi-resolution expansion is fully differentiable and therefore suitable for white box attacks. All transformations (downsampling, jittering, noise) maintain gradient flow, enabling complete white-box access.
>
> ### Linear Probe Training
> The probes were added on top of a frozen backbone model, no additional training was applied to the backbone while the probes were trained.
>
> Thank you for your detailed feedback. We hope we answered all your questions.

---

### Official Review · Reviewer_U5Sa · 2024-11-04

**Soundness:** 3
**Presentation:** 4
**Contribution:** 4
**Rating:** 8
**Confidence:** 3

**Summary:**

This paper addresses the challenge of adversarial robustness in deep neural networks. To tackle this problem, the authors of the paper propose a mechanism called CrossMax, based on Vickrey auction, to ensemble predictions from different layers given a multi-resolution input. The proposed method achieves adversarial robustness by leveraging the inherent robustness of intermediate layer predictions. The authors of the paper demonstrate that this method achieves significant adversarial accuracy on CIFAR-10 and CIFAR-100 without adversarial training. With the addition of adversarial training, the results can be further improved. Furthermore, the authors also explore the connection between adversarial robustness and the hierarchical nature of deep representations, showing that gradient-based attacks yield interpretable images of target classes. Additionally, they demonstrate that this approach enables controllable image generation using pre-trained classifiers and CLIP models.

**Strengths:**

- The paper is very well written and easy to follow. I enjoyed reading it.
- The proposed method is technically sound.
- The proposed idea is novel and generally applicable to a lot of different scenarios.
- The proposed method achieves impressive benchmark performance on various tasks.
- The authors of the paper conduct ample experiments to support their hypothesis.

**Weaknesses:**

- Most of the claims made in the paper seem to be empirical. It is not immediately clear how generalizable the conclusions are to other datasets and problems.
- The proposed multi-resolution approach can make the classifiers less computationally efficient.
- Hyperparameters on how to choose resolutions and number of resolutions can be hard to find/optimize.

**Questions:**

- I find the observation of using multi-resolution input to the classifier makes it much more adversarially robust interesting and somewhat surprising, since all the different resolutions are derived from the single-resolution image. Would it be possible at all to encoder this prior implicitly into the classifier model instead, so that we don't need to explicitly feed in N images of different resolution?
- Does the performance of the model get better as we feed in images of more and more resolutions?
- While I understanding the rationale of using Vickrey auction for ensembling, I wonder what inspired the use of it for self-ensembling? Does the same approach work for ensembling in general?

---

> ### Author Response · Authors · 2024-12-03
> **Response to reviewer U5Sa**
>
> Thank you for your positive review. We address your questions and concerns:
>
> ## Empirical Claims & Generalization
>
> We've now tested generalization to ImageNet, achieving promising preliminary results despite limited compute:
> - Multi-resolution backbone: 65% → 32% under AutoAttack (L∞ = 4/255)
> - CrossMax self-ensemble: 63% → 59% after attack
> - Comparable to current SOTA (59.64%)
>
> While preliminary (32 validation images), this indicates scalability beyond CIFAR datasets.
>
> ## Computational Efficiency
>
> The multi-resolution approach does increase computation, but only in a very limited way and with practical benefits:
>
> 1. In the backbone, the only architectural difference is the first convolution layer accepting as input 4x more channels than before. After that, all layers stay the same. This is a very small overhead compared to the standard ResNet152
>
> 2. For the self-ensemble, a selected layers have an additional linear projection (=a batch norm + a single affine layer) attached to them. In the implementations in the paper, we only use a selection of 8 such layers, inducing a limited computational overhead of 8 matrix multiplication in addition to the standard ResNet152
>
> 3. The preprocessing of a single image to a cascade for lower resolution version is very computationally efficient and does not pose a significant overhead.
>
> Overall the additional compute needed to run a multi-resolution self-ensemble is not large and not a major concern.
>
> ## Resolution Selection
>
> We have not studied resolution sensitivity in detail but found, by unguided experimentation, that a relatively small number of them sufficed. In the multi-resolution prior applied to generation, we see that even a set of resolutions corresponding to 2x multiples from the full resolution to the lowest possible (e.g. [224, 112, 56, 28, 14, 7]) was sufficient to generate interpretable images. We therefore believe a similar effect might be at play for the multi-resolution prior for defense.
>
> ## Implicit Multi-resolution Learning
>
> > “ explicitly feed in N images of different resolution”
>
> We actually found that feeding N images in doesn’t work, and that we needed to stack these versions along the channel dimension into a single 224x224x(3N) image and pass it through the network all at once.
>
>
> We tried a large number of ways of providing the model with the image at multiple resolutions, including passing them through multiple forward passes and aggregating their predictions at the end somehow. The only method that found to work as the one presented: producing a channel-wise stack of multiple resolutions, fed into the model **all at once** as channels of the same image. What makes this even more interesting is that this, on its own, wasn’t enough. We still need to train with a very low learning rate to get the accuracy as well as adversarial accuracy from such a model. With a high learning rate, we saw the accuracy being high, while adversarial accuracy keeping to around 0%. This to us suggests that it is indeed relatively difficult to make a model make use of such multiresolution input.
>
>
> ## Vickrey Auction in Self-ensembling
>
> The key insight was that intermediate layers make partially independent predictions under attack and that the layer the attack is not designed to specifically fool retains some ability to see the ground truth class. CrossMax's aggregation mechanism prevents any single layer from dominating, similar to how Vickrey auctions prevent bidder manipulation.
>
> We tested CrossMax for general ensemble aggregation:
> - 10 independently trained ResNet18s
> - L∞ = 8/255 AutoAttack
> - CrossMax improves robustness by 17x vs standard mean
> - Similar gains on other architectures, including Vision Transformers
>
> This suggests CrossMax's benefits extend beyond self-ensembling to general robust aggregation of even independently trained models.
>
> We will add these analyses to the final version to better support our method's broad applicability.

---

### Official Review · Reviewer_ircY · 2024-11-04

**Soundness:** 2
**Presentation:** 4
**Contribution:** 4
**Rating:** 5
**Confidence:** 4

**Summary:**

This paper proposes an innovative method for adversarial robustness in deep learning through multi-scale input processing and dynamic self-ensembling. By introducing multi-resolution input representations and CrossMax ensembling—a unique aggregation strategy that reduces susceptibility to adversarial perturbations—the authors effectively address key limitations in existing robustness approaches. This work is particularly notable for its ability to achieve competitive performance on CIFAR-10 and SOTA results on CIFAR-100 without relying on adversarial training, a common but computationally intensive technique.

**Strengths:**

The paper was well written and well structured. The flow of the paper was easy to follow. And I would like to thank the reviewers for their presentation.

I really liked that the idea of multi-scaling was inspired by the real human biology. I also enjoyed the cleaver trick leveraged in the CrossMax ensembling (and also in self-ensembling) to avoid fitting to the outlier predictions.

The results provided in the paper have significant value and are truly incredible. The idea is novel and the contributions are adequate.

**Weaknesses:**

The only weakness of the paper seems to be lack of evidence for adaptive attacks. I am not exactly sure if the following will be the best implementation, but just as an example the attacker could possibly optimize the input image with the objective of making a misclassification to happen across all layers in self-ensembled networks, or if the adaptive attack could optimize the input image across all resolutions, for white box attacks.

**Questions:**

1) As I previously mentioned, I think the paper could really benefit from some analysis regarding its resistance to some relative adaptive attack.

2) I also believe that since their method can adapt quite rapidly from a naturally trained model into an adversarially robust classifier, it would be a missed opportunity if it did not provide robustness results on ImageNet (or any other larger / more complex datasets).

3) In your setting, could the authors clarify if the attacker is aware of the parameters of the multi-resolution inputs? And also, for a single sample, is the process of generating different resolutions of the same image a randomized process or a fixed process for iterative PGD attacks? And also, how much the authors attribute the robustness to the randomization. If yes (and the process is randomized), could they provide similar results by fixing the random seed throughout the iterative generations to see if that effects the effectiveness of auto-attack.

4) On 336, why was the batch-norm turned off for the setting with training from the scratch?

5) Could the authors please enhance their representation of other methods? For instance,  mention the name of their method or name of the authors or some other representation that would give some sense of what the other baseline is.

6) Not a question, only a comment, the material covered in the appendix was really interesting and I really enjoyed them. I would be really happy to increase my score if the authors can convince me that the robusntess is preserved, even with adaptive attacks.

---

> ### Author Response · Authors · 2024-12-03
> **Response to Reviewer ircY**
>
> Thank you for your thoughtful and detailed review. We address your concerns and questions:
>
> ## 1. Adaptive Attacks
>
> During the rebuttal period, we implemented adaptive attacks specifically targeting our model's key components. Our approach:
>
> 1. Created a surrogate model removing CrossMax's max operations
> 2. Replaced median/top-k with simple mean aggregation
> 3. Ran PGD (simpler than AutoAttack to make sure we get as clean a signal as possible) attacks with increasing steps (20 to 640)
>
> Results on CIFAR-100 (L∞ = 8/255):
>
> | Model & Attack Details | PGD Steps |||||||
> |---|---|---|---|---|---|---|---|
> | | 20 | 40 | 80 | 160 | 320 | 640 |
> | Surrogate (mean, no max sub) | 52.5% | 45.1% | 31.6% | 24.1% | 22.7% | 22.3% |
> | Original (top-k) | 53.9% | 46.3% | 32.7% | 28.5% | 26.3% | 25.5% |
> | Original (median) | 53.1% | 45.8% | 32.9% | 26.3% | 24.1% | 22.5% |
>
> Key findings:
> - Attack transfer is effective, demonstrating the adaptive attacks’ strength
> - Performance stabilizes at ~25% despite exponential step increase
> - Successful attacks produce semantically meaningful changes (we would love to show you but can’t link images – we even ran a small human experiment during a talk at a university and the audience agreed that the “successfully attacked” image looked like the target label to them too! We will include it in the revised final version of the paper)
>
> ## 2. Preliminary ImageNet Results
>
> We agreed this would be valuable and have now run ImageNet evaluations. We were unfortunately limited to a single A100 in the end so could only perform a limited number of experiments. Using an ImageNet pretrained ResNet152 backbone and following our CIFAR-100 training procedure, we achieved promising results without any adversarial training:
>
> Multi-resolution setup:
> - Resolutions: [224, 112, 84, 56]
> - Noise level: 0.5
> - X-Y jitter: up to 42 pixels
>
> Results on ImageNet validation subset:
> - Multi-resolution backbone alone: 65% → 32% under AutoAttack (L∞ = 4/255)
> - CrossMax self-ensemble (layers [52,50,45,40,35,30], top-2): 63% → 59% after attack
>
> The RobustBench AutoAttack (with `rand` flag) shows as its detailed output:
> - Initial accuracy: 62.50%
> - Robust accuracy after APGD-CE: 46.88%
> - Robust accuracy after APGD-DLR: 43.75%
> - Final adversarial accuracy: 59.38% (due to some “successful” attacks being reversed by the random noise etc in the multi-resolution input preprocessing)
>
> (For comparison, the current best models get adversarial accuracy of 59.64% on the full ImageNet test set.)
>
> While preliminary (tested on first 32 validation images), these results indicate a promising robustness scaling to ImageNet. A complete evaluation and optimization would be valuable future work and we will try to add as much as we can to the final version.
>
>
> ## 3. Multi-resolution Implementation Details
>
> For each input image:
> - Resolutions: Fixed set {32×32, 16×16, 8×8, 4×4}
> - Randomization per forward pass:
>  - Random jitter (±3 pixels)
>  - Random noise (σ=0.1)
>  - Random contrast changes
>  - Random color-grayscale shifts
>
> The attacker has full white-box access to all parameters. The image transformations are all differentiable and do not block the white box access the attacker enjoys, making our model suitable for white-box evaluation. For PGD attacks, randomization occurs at each step with different seeds.
>
> ## 4. BatchNorm in Training from Scratch
>
> BatchNorm was disabled when training from scratch to maintain consistent statistics across the multi-resolution branches. With BatchNorm on, different resolutions developed significantly different statistics, harming performance. This is not a fundamental limitation, as BatchNorm can be re-enabled with appropriate normalization strategies across resolutions.
>
> ## 5. Method Comparisons
>
> We will enhance baseline descriptions in the final version by:
> - Adding method names
> - Including key architectural details
> - Providing clearer comparison tables
>
> We appreciate your thorough review, particularly regarding the appendix materials. The new adaptive attack results demonstrate our method's robustness even under targeted attacks while producing interpretable perturbations.
>
> Taken together, these results strengthen our paper's contributions by: (1) demonstrating robustness under adaptive attacks with asymptotic ~25% accuracy even with 640 PGD steps, (2) showing promising scalability to ImageNet with 59% adversarial accuracy (on RobustBench AutoAttack with a random small subset of images), and (3) clarifying technical implementation details. The interpretable nature of successful attacks against our model, verified through a small scale human evaluation, further supports our approach's soundness. We will incorporate these insights and clarifications in the final version.

---

### Public Comment · ~Matthias_Hein1 · 2024-11-19
**Robustness evaluation seems invalid**

Dear authors and reviewers,

we have read with great interest the paper "Ensemble everything everywhere: Multi-scale aggregation for adversarial robustness" in our reading group. Getting robustness without adversarial training would be great but the past showed that similar attempts to achieve strong robustness without adversarial training failed. In particular, there have been defenses involving randomization and changes in the activation functions/layers which made gradient-based attacks much harder but did not yield robust models.

We therefore were a bit sceptical and had a look at the evaluation using the published notebook (which the authors provide on their github page). First, of all we could reproduce the results. Then we modified the attack using APGD-CE for the undefended model:

- we replaced for the attack the top-3 with the mean (evaluation is still with the original model)

- we increased the number of EOT-iterations from 20 to 100

- we attack all samples even if they are classified incorrectly (as the defense is random, classification is not necessarily the same)

In this way we got for 32 points the robust accuracy for CIFAR100 from 62.5% down to 33.44% (these are averages at interference) and for CIFAR10 from 62.5% to 16.56%. More APGD iterations further reduce the robust accuracy. In particular, the results for CIFAR10 show that the model is not really robust and we are very sure that with more effort one can further reduce these numbers. The main problem is that their defense is essentially randomized and via the top-3 operation the gradients are sparsified leading to low progress with gradient-based attacks.

Thus we think that the claims of this paper are invalid. We informed the authors on October 30 about this and provided them with a modified notebook to reproduce our attack but got no answer so far.

Let me further add a few comments:
- the paper misses essential checks for adversarial defenses as described in Carlini et al, "On Evaluating Adversarial Robustness", e.g. to show that robust accuracy goes down to zero for large enough radius. But most importantly it does not try an adaptive attack against their defense as also requested by some reviewers. The attack we describe above is the most simple adaptive attack but one could think of better ones.

- it is unclear what the numbers of APGD-CE and APGD-DLR mean in Table 1. Note that they are all significantly lower than the reported adversarial accuracy. Our guess is that theses are the numbers for the samples used in the EOT attack. The gap between these numbers and the reported adversarial accuracy - just by introducing new randomness should already trigger some questions

- first experiments seem to suggest that the multi-resolution approach does not help much, one can get the same results by using the standard resolution but the reviewers can ask for an ablation about this

- the evaluation in Table 1 is only done for sample sizes ranging from 128 to 1024 whereas the test set of CIFAR10/100 has 10000 samples and this is the set on which RobustBench reports results (the last line denoted as SOTA) so the comparison is not fair

- the fact that robust classifiers lead to interpretable gradients and can be used for image synthesis is well known, see e.g.
  - Santurkar et al, Image synthesis with a single (robust) classifier, NeurIPS 2019
   - Augustin et al, Adversarial robustness on in-and out-distribution improves explainability, ECCV 2020
   - Boreiko et al, Sparse visual counterfactual explanations in image space, GCPR 2022

- the fact that averaging leads to some kind of robustness is also well known, see e.g.
  - Carmon, Y., Raghunathan, A., Schmidt, L., Duchi, J.C., Liang, P.: Unlabeled data improves adversarial robustness. In: NeurIPS (2019)

We are happy to discuss this and we can also provide the modified notebook so that everybody can reproduce the results of our attack (however we note that this would breach the anonymity of the authors, so we leave this to the AC if this is allowed or not)

Matthias Hein

---

> ### Author Response · Authors · 2024-12-03
> **A detailed reply to Prof Hein's comment [Part 1/3]**
>
> Dear Professor Hein,
>
> Thank you for your detailed comment. We will address specific points below, including the results for additional experiments and ablations we ran to clarify several points you raised. We must, however, strongly disagree with your sweeping conclusion that, as you say:
>
> > Thus we think that the claims of this paper are invalid.
>
> We hope that through our detailed reply you will revise your conclusion.
>
> Our step by step rebuttal of your detailed points follows:
>
> ## 1. Robust accuracy going to zero with large perturbation size
> You are concerned that we didn’t show explicitly that with large enough perturbation sizes, the adversarial accuracy goes to zero. We did perform this most basic check, but didn't include them in the paper, as it did not seem to add any value. However, as you suggested, here’s an additional experiment that we just ran, gradually increasing the $L_\mathrm{inf}$ of the RobustBench AutoAttack with the `rand` flag on, showing that our model’s robust accuracy decreases monotonically and all the way to 0% with the attack strength:
>
> The Table shows the Attack size L_inf out of 255 for the RobustBench AutoAttack with the `rand` flag on a subset of the CIFAR-100 test set and its effect on our Multi-resolution self-ensemble
>
> | Attack Size L_inf | Adversarial Accuracy |
> |-----------------|-------------------|
> | 8               | 59.38%           |
> | 12              | 40.62%           |
> | 16              | 31.25%           |
> | 32              | 3.12%            |
> | 64              | 3.12%            |
> | 128             | 0.0%             |
>
> As you can see, the increasing L_inf of the attack decreases adversarial accuracy quickly all the way to 0%, as is expected.
>
> ## 2. Adaptive attacks against our model
> You are correct to point out (as our reviewers also did) that we are missing an “adaptive” attack on our model. The word adaptive might mean e.g. adaptive learning rate in the attack etc, which is used as part of the AutoAttack already, but we believe that what is meant by it in this context is an **attack that is specifically tailored to our multi-resolution self-ensemble**.
>
> During the rebuttal period, we actually added new experiments to address this very issue. To make gradients as smooth as possible for the attack, we
> 1) removed the 2 max subtractions from cross-max, and
> 2) aggregated predictions from different layers using a simple mean instead of median or top_k.
> For the sake of simplicity, we used a standard Projected Gradient Descent (implemented in standard package as an attack = fb.attacks.LinfProjectedGradientDescentAttack(steps=attack_steps)), attacked this surrogate model, and then evaluate the attacked images on our original model, using their similarity for attack transfer.
>
> Our results for CIFAR-100 follow:
> PGD attack, steps = below (all using 8/255 L_inf size limit)
> | Description | Model details | 20 | 40 | 80 | 160 | 320 | 640 |
> |------------|---------------|-----|-----|-----|------|------|------|
> | The directly attacked surrogate model | No max subtractions, mean over layers | 52.5 | 45.1 | 31.6 | 24.1 | 22.7 | 22.3 |
> | The model we transfer the attacks to | 2x -max, top_k | 53.9 | 46.3 | 32.7 | 28.5 | 26.25 | 25.5 |
> | The model we transfer the attacks to | 2x -max, median | 53.1 | 45.8 | 32.9 | 26.3 | 24.1 | 22.5 |
>
>
> We do see transfer from the surrogate model to our model, demonstrating that we are indeed using a strong, suitable adaptive attack here. The accuracy on the directly attacked surrogate model is lower than on our original model that we transfer the attacked images to (as expected), and the gap increases with the number of PGD attack steps used. Notice that even though we are increasing the number of steps geometrically (2x each step), the “returns” to this increase are flatlining and there seems to be a ~25% asymptotic accuracy, demonstrating that **even against a specifically tailored attack, our model’s robust performance does not go to 0%, even when allowing the attacker to use an exponentially increasing number of iterations to design the attack**.
>
> An interesting point we see is that when we visualize the resulting perturbations that successfully changed the images’ classes, they are universally very smooth, non-noisy, and interpretable, suggesting that our model is genuinely robust, as you suggested with one of the references you cite. In fact the changed images often do end up looking like the target class. This to us suggests that the issue could be more with the brittleness of the CIFAR-100 dataset and the task of measuring adversarially robustness as a resistance to a class flip from the original class to **any other class**, rather than with the model itself, but to properly address this we would need to write a separate paper.

---

> > ### Author Response · Authors · 2024-12-03
> > **A detailed reply to Prof Hein's comment [Part 2/3]**
> >
> > ## 3. Explaining Table 1 numbers
> >
> > > “unclear what the numbers of APGD-CE and APGD-DLR mean in Table 1”
> >
> > When you run the RobustBench AutoAttack standardized evaluation for models with randomized components (=with the `rand` flag enabled), the model is attacked first by the APGD-CE attack and then the APGD-DLR attack. Starting from some initial accuracy, the evaluation first reports the accuracy remaining after the APGD-CE attack, and as a second number it reports the accuracy remaining after the AGPD-DLR attack. These are the two numbers reported in the table – they are exactly what the benchmark evaluation reported to us when we ran it on our model.
> >
> > This generates attacked images that the benchmark then passes to the model again and reports the corresponding accuracy as “adversarial accuracy”, as we do in the paper. Because the generated attacks are usually relatively brittle, it is not infrequent that some images that the benchmark thought were successfully misclassified get in fact correctly classified at the end. This is why the accuracy after the second attack (APGD-DLR) can be lower than the final number for “adversarial accuracy” the evaluator reports to us.
> >
> > We agree that this might mean that the benchmark needs rethinking and strengthening in the future, but
> > a) given that it is the standard benchmark to compare to other models, and
> > b) that it offers a `rand` flag for randomized defenses,
> > We did all we could to make sure we are comparing our model to others fairly. What you are suggesting might require a complete redesign of the benchmark and would, while a worthwhile endeavour, certainly be outside the scope of our paper focused on model building and methods exploration.
> >
> > ## 4. The effect of multi-resolution?
> >
> > > “first experiments seem to suggest that the multi-resolution approach does not help much”
> >
> > We’re not sure what you’re referring to. In fact we show that in terms of adversarial accuracy after the RobustBench AutoAttack, roughly ½ of the performance is due to the multi-resolution alone (we call this the backbone) and the other ½ due to the self-ensemble. We explicitly show this in e.g. Figure 6, both for CIFAR-10 and CIFAR-100. The purple bar is the performance of the multi-resolution backbone alone (omitting any intermediate layer ensembling; 25% for CIFAR-100 w/o adv. training), while the light blue/greenish bar is the multi-resolution self-ensemble = the combination of both the multi-resolution input as well as the aggregation of intermediate layer predictions (46% for CIFAR-100 w/o adv. training).
> >
> > ## 5. Evaluation on a subset of the test set
> >
> > > “the evaluation in Table 1 is only done for sample sizes ranging from 128 to 1024 whereas the test set of CIFAR10/100 has 10000 samples and this is the set on which RobustBench reports results (the last line denoted as SOTA) so the comparison is not fair”
> >
> > We evaluated our strongest models on ~1000 test set examples due to time and compute constraints – with the `rand` flag triggering test time augmentations and slowing everything down by a large multiplier, even this took >10 hours per model per evaluation. However, the key disagreement with your objection is that, statistically, it is a completely valid approach to take 1,024 images randomly chosen out of the 10,000, measure robustness on them, and report back the mean and standard deviation, as we do. CIFAR-100 is not extremely heterogeneous and especially not if we randomly shuffle its (image, label) tuples. We subdivide the e.g. 1,024 into groups of 32 (of course each image has an attack developed against it individually), and report the mean and the standard error of the 32 groups of 32, giving as a good handle on both the mean robust accuracy per image as well as its standard error, which is more than what is typically reported (= just a single number). For example, in Table 1 we report the adversarial accuracy of the multi-resolution self-ensemble without adversarial training as 46.29 ± 2.36, including both the mean and standard error. We agree that we should have such numbers for all estimates that do not use the full 10,000 test set samples – due to time constraints we opted to have the error bar estimates only for the best performing, highest stakes models in our Tables in order to compare them fairly with other people’s benchmarked models. In Table 2, we do the same for the lightly adversarially trained models, where we e.g. report 51.28 ± 1.95 for the same model type also on CIFAR-100.

---

> > > ### Author Response · Authors · 2024-12-03
> > > **A detailed reply to Prof Hein's comment [Part 3/3]**
> > >
> > > ## 6. Related work
> > > We are happy to cite the valuable references you provided. While there are indeed high-level similarities in outcomes (robust models producing interpretable gradients), our work introduces fundamentally different mechanisms and novel technical contributions. While Santurkar et al. (NeurIPS 2019) demonstrated that adversarially trained models can be used for image synthesis, our work introduces a fundamentally different mechanism – achieving interpretable gradients through multi-resolution input processing rather than through adversarial training. Unlike their approach which requires robust training first, our approach (when flipped around) also turns standard pre-trained classifiers and CLIP models into controllable generators without any additional training at all, simply by expressing perturbations as sums across different resolutions. Similarly, while Augustin et al. (ECCV 2020) and Boreiko et al. (GCPR 2022) explored connections between robustness and explainability, our work achieves these properties through novel architectural choices rather than training procedures.
> > >
> > > Regarding the connection to averaging methods, our self-ensemble aggregation approach is fundamentally different from the techniques in Carmon et al. (NeurIPS 2019). While they use unlabeled data and standard averaging to improve robustness, our method achieves strong results without requiring any additional data. Furthermore, we discover and utilize the partial layer-wise decorrelation of adversarial susceptibility – a previously unexplored phenomenon – that allows us to build robust self-ensembles from a single model. These results suggest our approach represents a novel direction for building robust and interpretable models, complementary to existing methods.
> > >
> > > —--------------------------
> > >
> > > Having addressed the specific technical concerns, we now turn to broader considerations about evaluation methodology and the scope of our contributions:
> > >
> > > ## General points
> > >
> > > In general, we took all precautions with our evaluation on the standard RobustBench AutoAttack benchmark, including using their `rand` flag for models using randomized defenses, precisely because we wanted to be able to compare our scores to other people’s results. We believe this comparison is fair, since all entries in the RobustBench leaderboard are evaluated using the same suite of attacks. While it is true that specifically crafted, adaptive attacks might decrease a model's adversarial accuracy further than a fixed, standardized benchmark, a clear advantage of a standard benchmark is to be able to compare the relative robustness of different models and approaches.
> > >
> > > In general, we do not claim that we solved adversarial robustness, but rather show a promising and novel path towards improving it. We demonstrate that the performance without adversarial training is better than previous SOTA on RobustBench; one could also use adversarial training on top of our method as we show their complementarity. We appreciate your valid concern about adaptive attacks. As a result, we ran additional adaptive attack evaluations on our model, and analyzed the resulting successful attacks by human inspection.
> > >
> > > A key point we would like to emphasize is that the Colab demonstration that you found (and that we can’t link here directly for the sake of anonymity) is only training a quick demonstration model that is substantially weaker than the ones that we used in the paper. Hence, the results may not be directly comparable. We should have emphasized this more in the demo.
> > >
> > > —--------------------------
> > >
> > > We would like to thank you for detailed feedback again and while we appreciate your comments, we have to disagree with your characterization that “the claims of this paper are invalid”. We hope that the additional details and experiments we provided here clarify some of the points of disagreement.
> > >
> > >
> > > Best wishes,
> > >
> > > Authors

---

> > > > ### Public Comment · ~Matthias_Hein1 · 2024-12-05
> > > >
> > > > I would like to thank the authors for their detailed response. I still think that the claims regarding adversarial robustness are invalid.
> > > >
> > > > - independently of us Jie Zhang, Kristina Nikolić, Nicholas Carlini, Florian Tramèr published the paper "Gradient Masking All-at-Once: Ensemble Everything Everywhere Is Not Robust",  https://arxiv.org/abs/2411.14834. The abstract says: "In this short note, we show that this defense is not robust to adversarial attack. We first show that the defense's randomness and ensembling method cause severe gradient masking. We then use standard adaptive attack techniques to reduce the defense's robust accuracy from 48% to 1% on CIFAR-100 and from 62% to 4% on CIFAR-10, under the ℓ∞-norm threat model with ε=8/255."
> > > >
> > > >   Based on our own evaluation and the stronger evaluation of this paper, this shows that the models are not robust.
> > > >
> > > >
> > > > - the authors suggest that RobustBench allows randomized defenses and that "....this might mean that the benchmark needs rethinking and strengthening in the future, but a) given that it is the standard benchmark to compare to other models, and b) that it offers a rand flag for randomized defenses"
> > > >
> > > >   I want to clarify that **RobustBench explicitly does not allow randomized defenses**. On the github page it says
> > > >
> > > >  "Evaluation of the robustness to Lp perturbations in general is not straightforward and requires adaptive attacks (Tramer et al.,(2020)). Thus, in order to establish a reliable standardized benchmark, we need to impose some restrictions on the defenses we consider. In particular, we accept only defenses that are (1) have in general non-zero gradients wrt the inputs, **(2) have a fully deterministic forward pass (i.e. no randomness)** that (3) does not have an optimization loop. Often, defenses that violate these 3 principles only make gradient-based attacks harder but do not substantially improve robustness (Carlini et al., (2019)) except those that can present concrete provable guarantees (e.g. Cohen et al., (2019))."
> > > >
> > > > This restriction excluding randomized defenses has a good reason as explained in more detail in the RobustBench paper: https://arxiv.org/abs/2010.09670.
> > > >
> > > > "To evaluate defenses with stochastic components, it is a common practice to combine standard gradient-based attacks with Expectation over Transformations [5]. While often effective it might be not sufficient, as shown by Tramèr et al. [142]. Moreover, the classification decision of randomized models may vary over different runs for the same input, hence even the definition of robust accuracy differs from that of deterministic networks. We note that randomization can be useful for improving robustness and deriving robustness certificates [82, 25], but it also introduces variance in the gradient estimators (both white- and black-box) making standard attacks much less effective"
> > > >
> > > > - the following claim of the authors:
> > > >   "We believe this comparison is fair, since all entries in the RobustBench leaderboard are evaluated using the same suite of attacks."
> > > > is not correct. **RobustBench explicitly allows and encourages adaptive attacks and orders the results according to the lowest reported robust accuracy (column "best known robust accuracy").**

---

### Meta-Review · Area_Chair_JASg · 2024-12-12

**Metareview:**

This paper proposes a defense against adversarial attacks that uses hierarchical self-ensembles which are augmented using jittering and noise. While the paper has received good initial reviews, the validity of the approach has been publicly challenged during the discussion. In particular, the evaluation on RobustBench, on which the paper claims to achieve state-of-the-art results, is not entirely valid. As clarified by the authors during the rebuttal, the proposed approach remains fully differentiable through the noise and jittering operations. Therefore, the attacks executed in RobustBench succeed to some extent. Yet, the SotA on RobustBench is compared on deterministic models - the randomization within the proposed method switches the paradigm so that the adversarial robustness between the referred methods is not directly comparable.
Following this discussion, and in the context of the positive ratings given by the reviewers, I have carefully read the paper. I agree with the reviewers that the paper is well written and motivated, and that the measured increase in robustness is promising - even if the results on RobustBench are not comparable to previous results reported there. However the biggest flaws of the paper are to be seen in two aspects: (1) The limited novelty of the approach and (2) the missing comparison to previous adversarial defenses the incorporate randomization. In particular, there are several previous works stating that self ensembles and hierarchical representations improve adversarial robustness, in particular when considered with additional noise.
Examples would be:
- Liu, X., M. Cheng, H. Zhang, and C.-J. Hsieh, Towards robust neural networks via random self-ensemble, ECCV, 2018.
- X. Wang, S. Wang, P.Y. Chen, X. Lin, and P. Chin, Advms: A multi-source multi-cost defense against adversarial attacks. In ICASSP, 2020.
- X. Wang, S. Wang, P.Y. Chen, Y. Wang,B. Kulis, X. Lin, and P. Chin. Protecting neural networks with hierarchical random switching: Towards better robustness-accuracy trade-off for stochastic defenses. IJCAI 2019.

The delineation to these previous works is missing in the submitted manuscript,  a proper evaluation of the proposed method in the paradigm of defenses through randomization is also missing. Therefore, even though the manuscript is well written and the proposed technique might have promise, it is not ready for acceptance.

**Additional Comments On Reviewer Discussion:**

After the public comment, the reviewers did not engage in discussion among themselves. The authors have provided their rebuttal on the last possible day, also leaving no time for direct interaction. Direct discussion with the AC with one of the very positive reviewers during the reviewer-AC discussion phase showed that this review is of low confidence - which might also result from the public discussion. The other two positive reviewers were not responsive during the entire discussion phase - leaving the paper without champion in the end.

---

### Decision · Program_Chairs · 2025-01-22

Reject